# Loss of the transcription factor MAFB limits β-cell derivation from human PSCs

Ronan Russell[1], Phichitpol P. Carnese[1], Thomas G. Hennings[1], Emily M. Walker [2], Holger A. Russ [1,3], Jennifer S. Liu[1], Simone Giacometti[1], Roland Stein[2] & Matthias Hebrok [1✉]

Next generation sequencing studies have highlighted discrepancies in β-cells which exist between mice and men. Numerous reports have identified MAF BZIP Transcription Factor B (MAFB) to be present in human β-cells postnatally, while its expression is restricted to embryonic and neo-natal β-cells in mice. Using CRISPR/Cas9-mediated gene editing, coupled with endocrine cell differentiation strategies, we dissect the contribution of MAFB to β-cell development and function specifically in humans. Here we report that MAFB knockout hPSCs have normal pancreatic differentiation capacity up to the progenitor stage, but favor somatostatin- and pancreatic polypeptide–positive cells at the expense of insulin- and glucagon-producing cells during endocrine cell development. Our results describe a requirement for MAFB late in the human pancreatic developmental program and identify it as a distinguishing transcription factor within islet cell subtype specification. We propose that hPSCs represent a powerful tool to model human pancreatic endocrine development and associated disease pathophysiology.

[1] UCSF Diabetes Center, University of California San Francisco, San Francisco, CA 94143, USA. [2] Department of Molecular Physiology and Biophysics, Vanderbilt University, Nashville, TN 37232, USA. [3] Present address: Barbara Davis Center for Diabetes, School of Medicine, University of Colorado Denver, Aurora, CO 80045, USA. ✉email: Matthias.Hebrok@ucsf.edu

dentifying key factors that mediate islet cell function is critical to understanding Type 1 and Type 2 Diabetes Mellitus (T1D, T2D) disease processes and to moving forward with cell-based interventions in humans[1]. Notably, the advent of in-depth sequencing studies has elucidated alternative transcription factor activity among closely related islet cell subtypes[2,3] and additionally, uncovered numerous examples of heterogeneity within healthy β-cells[4]. It has long been recognized that differences in cellular composition and architecture exist between human and mice adult islets, although the precise reasons and implications for such disparity have remained elusive. Numerous recent single-cell RNA-sequencing studies have reported a discrepancy in the expression of a number of factors, highlighting potential species-specific regulatory differences among rodent and human β-cells/islet counterparts[3,5].

One such factor is MAFB, which has been identified in numerous studies to be expressed within adult human β-cells, while being absent from the murine counterparts. MAFB is a member of the large Maf family of transcription factors that share highly similar basic region/leucine zipper DNA binding motifs and N-terminal activation domains. MAFB is expressed in developing pancreatic α- and β-cells in the mouse, while its expression becomes restricted to α-cells in the adult[6–8] and MAFB knockout (KO) in the mouse pancreas merely delays β-cell maturation for a short period of time[9,10]. MAFB binds to the MAF responsive element (MARE) expressed within critical β-cell genes including PDX1 and INS. Kim and colleagues recently showed that MAFB mRNA is expressed at robust levels both in α- and β-cells in humans with no significant fluctuations in levels over age[11], a finding that has since been demonstrated at protein level[12]. Building on this, single-cell RNA-sequencing studies have identified that MAFB is robustly expressed in human adult β-cells as well as α- and δ-cells[2,3,5,13–16]. This is a notable difference to the related factor MAFA, a β-cell specific transcription factor that also binds to the insulin MARE site, activates insulin gene transcription[7], and has been reported as a critical factor in adult β-cells across species. MAFA expression increases in an age-dependent manner in both human and mouse β-cells and is the predominant isoform expressed in functional mouse β-cells[12], although it is not robustly expressed in human β-cells until after ~10 years of age[3,11]. Together, these data suggest a potential evolutionary discrepancy in the role of MAFB in the pancreatic islet compartment and the use of MAFA as a marker of human β-cell function has been questioned in recent studies[17].

Directed differentiation of human pluripotent stem cells (hPSCs) into insulin producing β-like cells provides a powerful platform to investigate human biology. This approach has been shown to effectively recapitulate the sequential stages of islet cell development, in which hPSCs transition through a definitive endoderm (DE) stage, multipotent pancreatic progenitor (PP) stage, and toward β-like cells. By combining this strategy with genome editing approaches such as CRISPR/Cas9 (clustered regularly interspaced short palindromic repeats), it is possible to dissect the specific contribution of pancreatic transcription factors at precise stages in a human-specific context[18]. While much effort has been spent focusing on optimizing directed differentiation of human cell types for disease modeling or the phenotypes associated with known genetic susceptibility genes[19,20], the underlying complex biological processes which established protocols recapitulate also provides a unique platform for investigating the normal pancreatic developmental program as shown for PAX4 and ARX as well pan-pancreatic lineage determinants such as PDX1[18,21,22]. These reports have demonstrated the feasibility of this approach and verified in a human context, the precise role of established transcription factors in β-cell biology.

In the present study, we generate isogenic MAFB loss of function hPSCs to dissect its contribution in human β-cell development, using a stepwise differentiation protocol. MAFB loss of function cells display no major defects in the ability to specify early stages of pancreatic development. However, MAFB null cells have an increased propensity to generate somatostatin-(δ) and pancreatic polypeptide- (γ) producing cells at the expense of glucagon (α) and insulin (β) producing cells. Our results identify an important role of MAFB within endocrine cell-subtype specification decisions and demonstrate that it is a critical transcription factor for the generation of glucose-responsive β-cells from hPSCs. This essential role of MAFB in human β-cell identity is considerably different from what was concluded in genetically engineered mouse models (GEMMs). These findings suggest that species-specific differences in transcription factor activity may exist between mice and men and promote the use of human developmental models such as directed differentiation strategies to enhance our understanding of human biology.

## Results

**MAFB is expressed during β-like cell differentiation**. We employed our previously published β-cell directed differentiation protocol with modifications (Fig. 1a, Supplementary Tables 1 and 2)[23,24]. Utilizing the INS-GFP reporter hESC line, in which GFP is knocked into one endogenous locus of the insulin gene[25], this protocol routinely generates >90% definitive endoderm (DE) cells co-expressing SOX17 and FOXA2, >70% PP cells co-expressing PDX1 and NKX6.1, and 20–50% mono-hormonal β-like cells (β-like) co-expressing C-peptide and NKX6.1 as assessed by flow cytometry (FC) and immunofluorescence (IF) staining (Fig. 1b, c, Supplementary Fig. 1a). First, we examined the expression of MAFB over the time course of differentiation using qPCR and western blot analysis. mRNA levels of MAFB are first detected at the PP stage and increase in levels at the β-like stage (Fig. 1d). MAFB protein was principally detected at the β-like stage, indicating that robust MAFB expression coincides with the onset of hormone expression (Fig. 1e)[7].

In order to examine the role of MAFB in the differentiation of β-cells, we utilized an inducible (i)CRISPR gene editing system[26] in the INS-GFP cell line along with three individual guide RNAs (gRNAs) (Fig. 1f, Supplementary Table 3). Three independent MAFB heterozygous and homozygous clones with frameshift mutations in the transactivation domain, leading to disruption of the functional full-length protein, were established. Isogenic clonal cell lines which lacked genomic editing were employed as controls. All clonal hESC lines grew as expected compared with the parental control line, suggesting the genomic editing and sub-cloning procedure did not compromise pluripotency. To verify the genotypes, cell lines were assayed via targeted sequencing of the genome-edited locus at the pluripotency stage, as well as by western blot and IF analysis for MAFB protein at the β-like cell stage (Fig. 1g–h, Supplementary Fig. 1b). Mono-allelic and bi-allelic MAFB clonal lines harboring frameshift mutations will be referred to as +/− and −/−, respectively, throughout the text and specific indel sequence information for each clone is outlined in Supplementary Tables 4 and 5.

**Differentiation of MAFB KO hESCs to pancreatic β-like cells**. To control for potential off-target genomic editing, three individual MAFB+/− and −/− cell lines were differentiated in parallel to an isogenic MAFB+/+ subline as well as the parental INS-GFP line. Results are pooled by genotype in main figures and throughout the text unless otherwise stated and results of individual clones are presented in Supplementary Fig. 2a–d. All cell lines differentiated into DE cells co-expressing SOX17 and

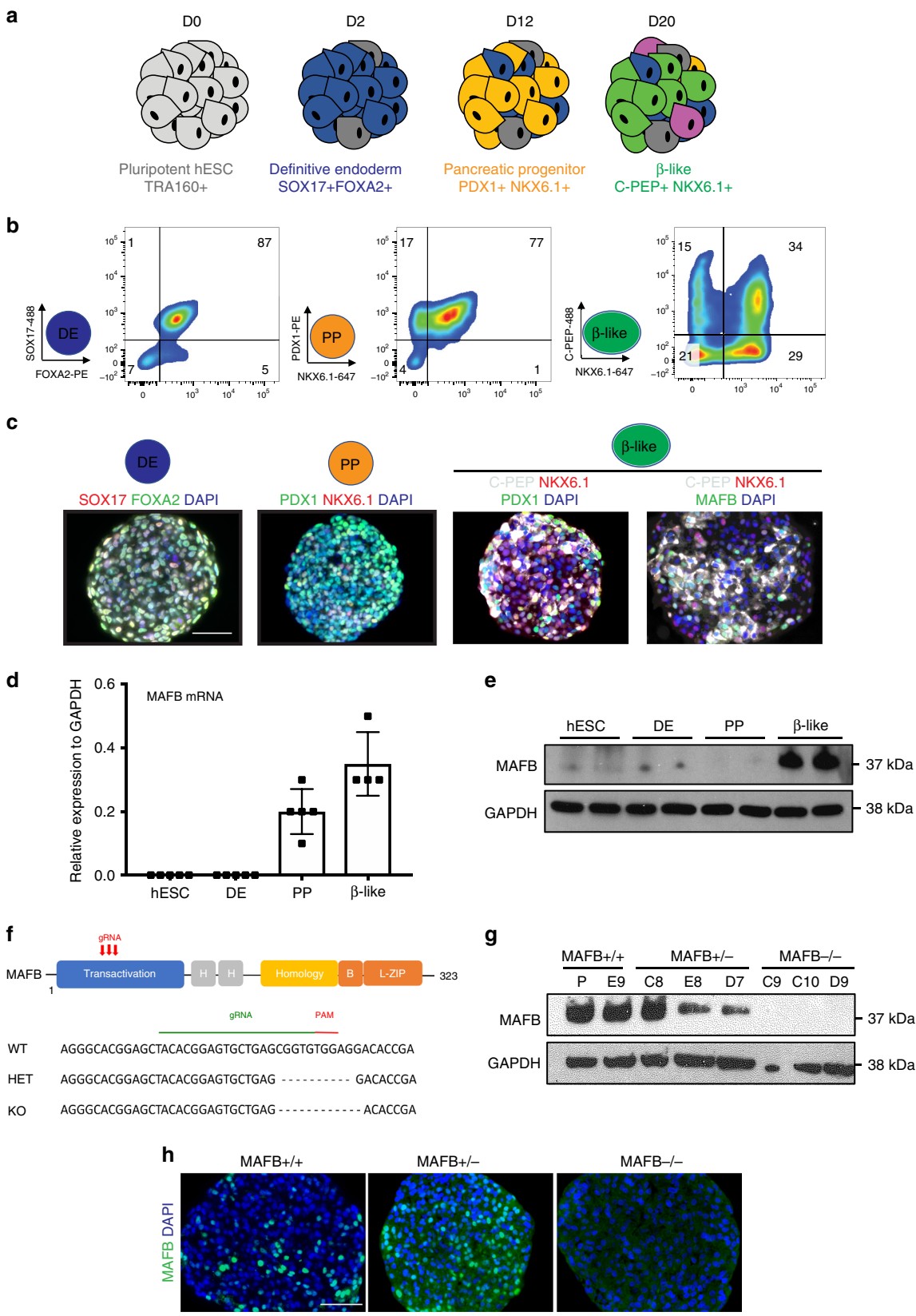

FOXA2, as well as PP cells co-expressing PDX1 and NKX6.1 with equal efficiencies as assessed by FC and IF analysis (Fig. 2a, b). However, further differentiation to the β-like stage revealed significant deficiencies of both MAFB+/− and −/− cell lines to generate INS-GFP+ cells compared with WT controls (35 ± 3.3% vs. 23 ± 5.6% vs. 4.4 ± 4.2%) (Fig. 2c). Moreover, loss of either one

or both alleles of MAFB led to a significant decrease of C-PEP + NKX6.1+ double-positive cells (24 ± 2.8% vs. 15 ± 5.7% vs. 3.7 ± 4.2%) (Fig. 2d).

We also observed an overall decrease in the levels of NKX6.1 in the MAFB−/− cells when compared to controls (62 ± 6.2% vs. 58 ± 11% vs. 42 ± 13%); a transcription factor critical for β-cell

**Fig. 1 MAFB is expressed in human β-cell counterparts and generation of MAFB KO hESCs. a** Schematic outlining the differentiation protocol to generate β-like cells from hESCs. The key lineage markers are outlined at each stage. Chemicals and durations for each differentiation stage are indicated in the Methods section. d, day(s); hESC, undifferentiated hESC stage; DE, definitive endoderm stage SOX17 + FOXA2+; PP, pancreatic progenitor stage PDX1 + NKX6.1+; β, β-like cell stage. **b** Representative FC plots outlining a typical differentiation of cells through the DE, PP, and β-like cell stages using the indicated markers. **c** Representative IF images from four independent experiments at indicated stages of differentiation depicting SOX17 and FOXA2, PDX1 and NKX6.1, C-PEP co-staining with PDX1 and NKX6.1, and MAFB co-staining with C-PEP and NKX6.1. DAPI indicates nuclear staining. Scale bars, 50 μm. **d** MAFB mRNA expression during β-cell differentiation as shown via qPCR. Each data point represents an independent biological sample. **e** Representative western blot from three independent experiments showing MAFB protein levels during β-cell differentiation at the indicated stages. **f** Targeted disruption of MAFB in hESCs. Targeting strategy for CRISPR-mediated gene editing of MAFB. **g** Representative western blot analysis of MAFB in targeted clones at the β-cell of differentiation from three independent experiments. Sizes of molecular weight markers are shown on the right. Similar results were obtained using two independent antibodies. **h** Representative IF images from three independent experiments at the β-cell stage of differentiation depicting MAFB expression in MAFB +/+, +/−, and −/− cells. DAPI indicates nuclear staining. Scale bars, 50 μm.

identity[27]. A similar pattern was also observed via IF analysis of the respective stages and live imaging of GFP+ cells (Fig. 2e, Supplementary Fig. 2e, 3a–c). In addition, we found that the mean fluorescence intensity (MFI) of GFP was significantly lower in MAFB+/− and −/− cells compared to controls, indicating a lower amount of INS on a per cell basis (Supplementary Fig. 2f, g). Taken together, these results establish a key role of MAFB in promoting β-cell formation.

**MAFB is required for formation of glucose-responsive β-cells.** To further investigate the functional consequences of MAFB loss, we transplanted MAFB+/+ and −/− PP and β-like cell stages under the kidney capsules of immunocompromised NOD SCID (severe combined immunodeficiency) beige mice (Jackson Laboratories) (Fig. 3a). In vivo engraftment of β-like cells coupled with glucose-stimulated insulin secretion (GSIS) assays is an established test of their competence[28]. As the phenotype of each individual genotype-matched clone was the same during in vitro differentiations, a single clone of MAFB+/+ and −/− was tested in vivo.

Due to pronounced growth of the graft, the mice which received PP cells had to be sacrificed by 8 weeks post-transplantation. Removal of the graft and subsequent IF analysis revealed no C-PEP positive cells within the graft of either MAFB+/+ or −/− cells, indicating that PP stage cells are not yet sufficiently specified toward the endocrine cell lineage.

In the mice which received the more differentiated β-like cells, circulating human C-peptide was readily detectable in the serum of MAFB+/+ recipient mice at 5 weeks post-transplantation. Upon glucose challenge, MAFB+/+ cells demonstrated a robust and significant increase in human C-peptide levels ($29 \pm 5.2$ pg/ml vs. $157 \pm 37$ pg/ml) (Fig. 3b, S2H). In contrast, MAFB−/− transplanted cells gave no response to glucose challenge, and circulating human C-peptide could not be detected in the serum before glucose challenge (Fig. 3b). This is in line with in vitro assessments whereby MAFB+/− and −/− cells had significantly less C-peptide content than MAFB+/+ counterparts at the β-like cell stage (Supplementary Fig. 1i). We next sought to look at the morphology of the grafts to determine if there were small clusters of insulin producing cells that may evade the limit of detection through secretion assays. IF analysis of the grafts complemented these findings whereby the frequency of C-PEP + NKX6.1 + PDX1+ cells identified was much higher in grafts derived from MAFB+/+ transplanted cells than in the MAFB−/− grafts, which had high levels of PDX1+ cells, low numbers of NKX6.1+ cells, and no detectable C-PEP positive cells (Fig. 3c, Supplementary Fig. 4a). These data suggest that the residual MAFB−/− β-like cell in vitro are non-functional even after in vivo maturation promoting conditions. These findings contrast with results derived from mouse studies in which β-cell maturation was merely slightly

delayed due to MAFB loss and where β-cell numbers recovered to normal postnatally[9].

**RNA-seq analysis of MAFB KO cells.** As there were no obvious phenotypic deficiencies within the MAFB−/− cells up to the PP stage of differentiation, but a significant reduction in the number of β-like cells formed, we investigated potential transcriptional differences via single-cell RNA sequencing (scRNA-seq). This approach provides the ability to address the issue of cellular heterogeneity within the differentiating cells that may affect the propensity to differentiate into β-like cells and has advantages as the identity of the cell source of a given RNA transcript is maintained unlike in bulk RNA-seq[29–31].

We analyzed the PP and β-like cell stages, and recovered 578 and 1126 cells for MAFB+/+ and 564 and 2387 cells for MAFB−/− cells, respectively. The Seurat pipeline[32,33] was utilized to provide unbiased transcriptome clustering of individual cells from MAFB+/+ and −/− cells at the indicated stages and the data are shown by uniform manifold approximation and projection (UMAP) plots (Supplementary Fig. 5a and 6b). We next performed integration of the MAFB+/+ and −/− samples at their respective stages, a method that facilitates identification of shared populations across samples and downstream comparative analysis[32]. First, we identified a strong overlap of cell transcriptomes at the PP stage of differentiation (Supplementary Fig. 5b). Moreover, there was a broad level of expression of PDX1 and NKX6.1 in both MAFB+/+ and −/− samples and differentially expressed gene (DEG) analysis revealed a highly similar transcriptome between these two samples, whereby no genes were differentially expressed outside the set thresholds (Supplementary Fig. 5c). In line with this, bulk mRNA analysis at the PP cell stage revealed no significant changes between MAFB+/+ and −/− cells in a variety of pan-pancreatic (PDX1, PTF1A, SOX9, and HNF6) or pro-endocrine genes (NGN3, NKX6.1, NKX2.2, and NEUROD1) (Supplementary Fig. 5d, e). These data indicate MAFB is required downstream of this stage, in line with our observation that MAFB protein is not detectable at the PP stage (Fig. 1d).

We next identified 10 independent clusters within integrated MAFB+/+ and −/− cells from the β-like cell stages and subsequently annotated the clusters according to the expression of well-known marker genes that have been shown to have an established role in pancreas development/function (Fig. 4a). These clusters were easily distinguishable using the pan-endocrine marker CHGA (Supplementary Fig. 6a) and markers including Dlk1, MEST, HES1, FEV and NGN3[34,35] and classical hormones including INS, GCG, PPY, and SST[36].

Applying the same DEG analysis criteria as above, we identified 16 up- and 26 downregulated genes in the MAFB−/− cells compared with controls (Supplementary Table 6). MAFB−/− cells presented strong upregulation of the pancreatic hormones

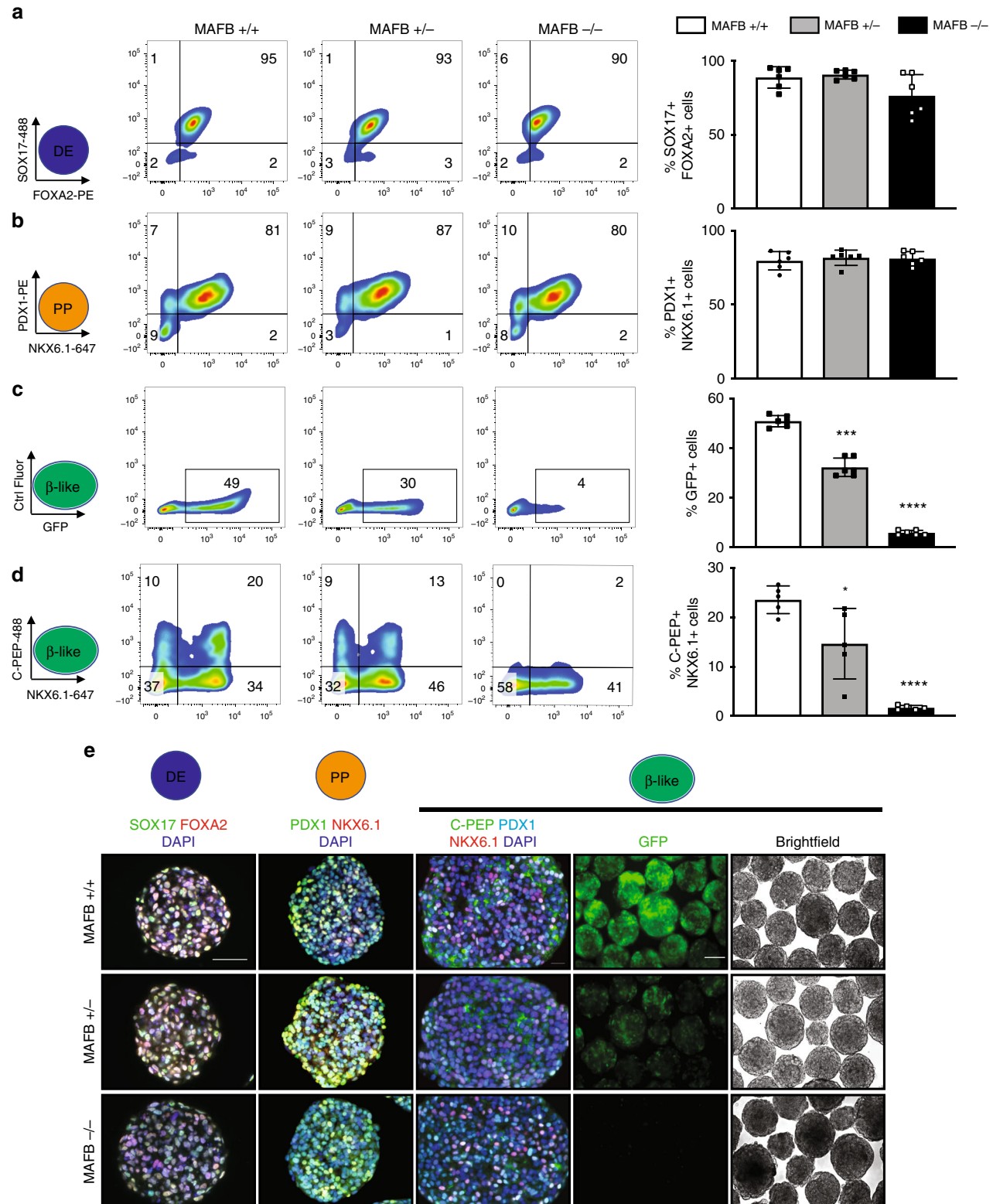

**Fig. 2 Loss of MAFB limits β-cell derivation. a** Representative FC plots depicting the percentages of SOX17 + FOXA2+ cells and quantification at the DE stage ($n = 6$, biologically independent samples). **b** Representative FC plots depicting the percentages of PDX1 + NKX6.1+ cells and quantification at the PP stage ($n = 6$, biologically independent samples). **c** Representative FC plots depicting the percentages of GFP+ cells and quantification at the β-cell stage ($n = 6$, biologically independent samples). **d** Representative FC plots depicting the percentages of C-PEP + NKX6.1+ cells and quantification at the β-cell stage ($n = 6$, biologically independent samples). **e** Representative IF images from six independent experiments at the indicated stages of differentiation depicting SOX17 and FOXA2, PDX1 and NKX6.1, and C-PEP co-staining with PDX1 and NKX6.1. Scale bars, 50 μm. GFP and Brightfield images from live in vitro cell cultures. Scale bars, 100 μm. $P$ values by one-way ANOVA followed by Dunnett's multiple comparisons test were *$P < 0.05$, **$P < 0.01$, ***$P < 0.001$, ****$P < 0.0001$. Data are presented as individual biological replicates and represent the mean ± SD.

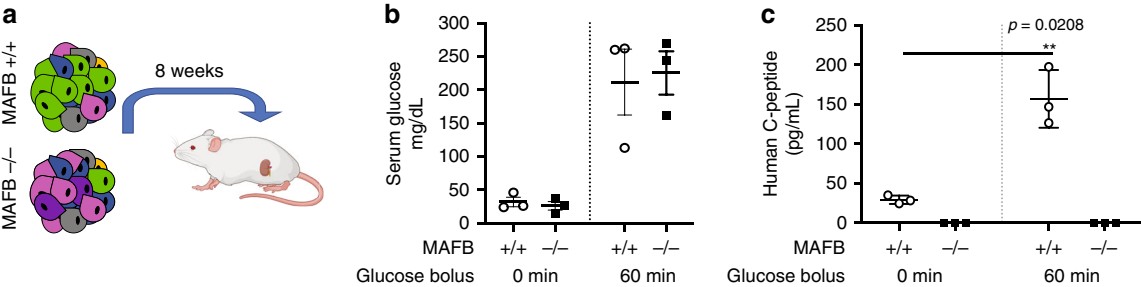

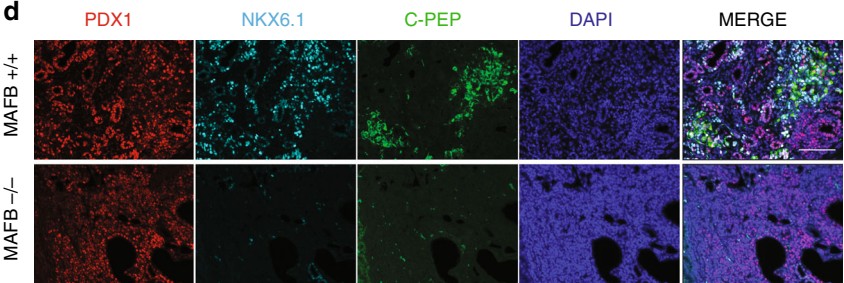

**Fig. 3 MAFB is required to generate functional β-cells. a** Schematic created with Biorender.com outlining the transplantation strategy to test the functionality of MAFB+/+ and −/− β-like cells under the kidney capsule of NSG mice. **b** Serum glucose measurements from glucose-stimulated insulin secretion (GSIS) experiments at 5 weeks post-transplantation (n = 3 independent mice). **c** GSIS experiment at 5 weeks post-transplantation. Experiments were performed one time with three independent NSG mice (n = 3) per group. P values by paired two-tailed t-test. **d** Representative IF images from six independent animals (3 MAFB+/+ and 3 MAFB−/−) for C-PEP co-staining with PDX1 and NKX6.1 of grafts removed from kidney capsules 8 weeks post-transplantation. Scale bars, 50 μm. Data are presented as individual biological replicates and represent the mean ± SEM.

*SST*, *PPY*, *GAST*, and *PYY* as well as the ISG15 gene and downregulation of the hormones *INS*, *GCG* and genes including *ACVR1C*, *TTR*, and *CRYBA2*. An overview of the distribution of gene changes is outlined in UMAP plots for MAFB+/+ and −/− samples in Supplementary Fig. 6c, d. In addition, an overview of gene transcriptional changes associated with each of the major hormone-producing cell clusters in MAFB+/+ vs. −/− cells is outlined in Supplementary Tables 7–9.

We next confirmed this remarkable shift in islet hormones via quantitative mRNA analysis. MAFB−/− cells had reduced levels of *INS* as expected, with a concomitant large increase in the levels of *SST* and *PPY*. *GCG* transcript levels had a trend toward lower levels, while those of *GHRL* were not significantly changed (Fig. 4c). Similar results were obtained for all clones as outlined in Supplementary Fig. 7b. We also assessed the expression levels of *CHGA* and major lineage specifying genes including *PAX4*, *PAX6*, and *ARX* which had no appreciable differences in MAFB−/− cells compared with controls. However, we did observe an increase in the levels of *HHEX*, a δ-cell marker, while the β-cell maturation factor *MAFA*[37] and the β-cell de-differentiation marker *ALDH1A3*[38] had a trend toward lower expression in MAFB−/− compared with controls (Supplementary Fig. 7b, c).

To determine whether the observed gene changes were an artefact of this particular in vitro system, we compared MAFB chromatin immunoprecipitation sequencing (ChIP-seq) data from published adult human islets[39,40] and EndoC-BH2 cells with our list of top DEGs (Supplementary Fig. 8). We observed that MAFB bound in close proximity to the promoter region of *INS*, *TTR*, and *CRYBA2* as well as within the *ACVR1C* gene, while there was a MAFB binding peak upstream of the *GAST* promoter region in human islets (Supplementary Fig. 8, upper panel). The prevalence of MAFB binding peaks within genes from human islets suggests that our differentiation protocol acutely reflects development of bone fide endocrine cell populations. Moreover,

by comparing these datasets with the EndoC-BH2 cell line, we have evidence that many of these effects are likely to be β-cell specific (Supplementary Fig. 8, lower panel). Notably, the overlap with ChIP-seq data was higher with genes whose expression was decreased in the absence of MAFB, suggesting that it primarily functions as a transcriptional activator in endocrine cells. One limitation of this hypothesis is the lack of purified cell lines from other hormone-producing cells such as α-, δ-, and γ-cells which are expressed at much lower frequencies in human islets compared to β-cells[41].

Notably, the hormones *GAST* and *PYY* have been reported to be expressed in the developing mouse pancreas and also in hPSC β-cell differentiation protocols, although they are absent from mature β-cells except in type 1 diabetes when GAST becomes upregulated[42–45]. Together, these data advocate that MAFB acts as a late-stage endocrine cell-fate rheostat in humans, in a similar manner to the α- and β-cell lineage determinants ARX and PAX4, respectively[21,22].

**MAFB regulates endocrine cell lineage commitment.** To further expand these observations, we performed FC analysis of the β-like cell stage for the pan-endocrine marker CHGA to assess the percentage of total endocrine cells present. There was no significant difference in the levels of CHGA+ cells in the MAFB−/− cells (77 ± 3.4% vs. 75 ± 8.5% vs. 71 ± 5.9%) (Fig. 4d, Supplementary Fig. 9a) indicating that MAFB loss alone is sufficient to compromise the composition of the endocrine cell compartment without affecting total numbers of endocrine cells. Similarly, we did not detect any difference in the levels of NGN3 mRNA levels between MAFB+/+ and −/− cells, indicating that equivalent numbers of endocrine progenitor cells were specified. We also did not detect a significant change in the rate of proliferation as assessed by KI-67 FC between MAFB+/+ and −/− cells either at the PP stage (13 ± 9% vs. 11 ± 6.2%) or the β-like cell stage (4.6 ± 1.6% vs. 5.7 ± 2.4%) (Supplementary Fig. 9e, f).

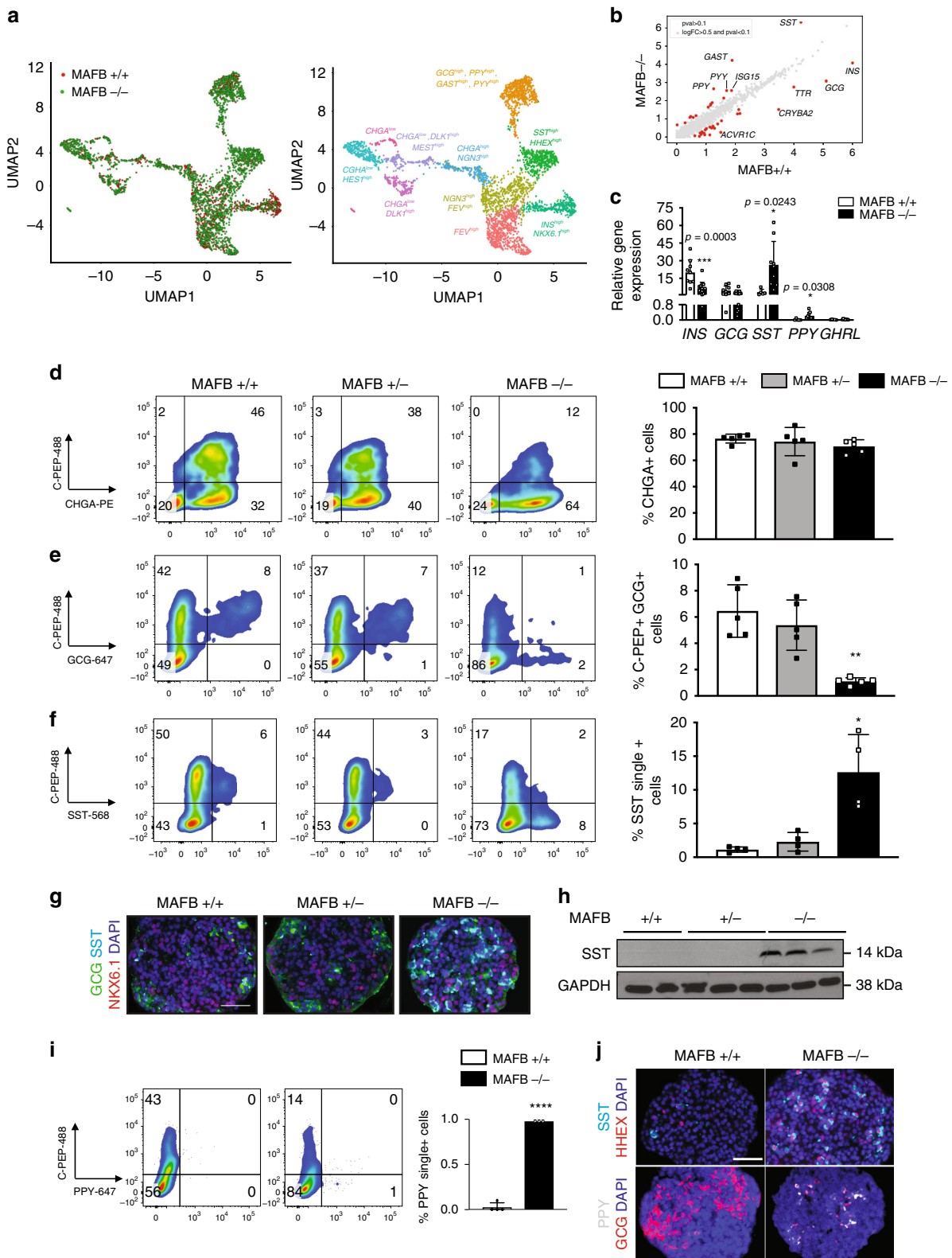

However, in line with the mRNA data, we found a significant increase in the levels of SST single positive cells in MAFB−/− cells (1 ± 0.5% vs. 2 ± 1% vs. 13 + 6%) as well as a dose-dependent reduction in the levels of GCG (6.5 ± 2% vs. 5 ± 1.8% vs. 1.2 ± 0.46%) (Fig. 4e, f). An overview of clonal cell lines is provided in Supplementary Fig. 9a–d. IF analysis confirmed SST was strongly upregulated in the MAFB−/− cells and were mostly single

hormone positive in contrast to the majority of SST+ cells within controls (Fig. 4g, Supplementary Fig. 10a). Western blotting revealed that the pancreatic specific 14 kDa peptide of SST was strongly upregulated in the MAFB−/− cells (Fig. 4h). Further-more, we also identified a significant increase in the levels of PPY via FC analysis, representing a ~10-fold increase in this rare cell type in MAFB−/− cells compared with controls (0.1 ± 0.05% vs.

**Fig. 4 MAFB controls pancreatic endocrine cell lineage specification. a** UMAP projections of integrated MAFB+/+ and −/− transcriptomes, color-coded by genotype (left) and cell populations (right). **b** Scatter plot showing the average gene expression (log scale) for MAFB+/+ and −/− cells at the β-like cell stage. Differentially expressed genes (log fold change >0.5, adjusted $P$ value <0.1) between MAFB+/+ and −/− cells are shown in red. Significance was calculated using the MAST test and $P$ values were adjusted for multiple testing using the Benjamini–Hochberg method. The top FIVE up- and downregulated genes are indicated. **c** qPCR analysis showing mRNA levels of islet hormones (*INS, GCG, SST, PPY,* and *GHRL*) at the β-cell stage by. $P$ values by unpaired two-tailed $t$-test were *$P < 0.05$ and ***$P < 0.001$. **d–f** Representative FC plots depicting **d** C-PEP+ and CHGA+ cells ($n = 5$, biologically independent samples), **e** C-PEP+ and GCG+ cells ($n = 5$, biologically independent samples), and **f** C-PEP+ and SST+ cells ($n = 4$, biologically independent samples) and respective quantification at the β-like cell stage. **g** Representative IF images from four independent experiments at the β-cell stage of differentiation depicting GCG, SST, NKX6.1, and DAPI. Scale bars, 50 μm. **h** Western blotting for SST protein (14 kDa) expression at the β-cell stage. GAPDH (38 kDa) was used as a loading control ($n = 2$ biologically independent MAFB+/+ samples and $n = 3$, biologically independent samples for MAFB + /− and −/−). **i** Representative FC plots depicting the percentages of C-PEP + PPY+ cells and quantification at the β-like cell stage ($n = 4$ for MAFB+/+ and $n = 3$ for MAFB−/−, biologically independent samples). $P$ values by unpaired two-tailed $t$-test were ****$P < 0.0001$. **j** Representative IF images from four independent experiments depicting SST and HHEX (upper panel) and PPY and GCG (lower panel) and DAPI. Scale bars, 50 μm. $P$ values by one-way ANOVA followed by Dunnett's multiple comparisons test were *$P < 0.05$, **$P < 0.01$, ***$P < 0.001$, ****$P < 0.0001$. Data are presented as individual biological replicates, representing the mean ± SD. N numbers indicated in Supplementary Fig. 2a, b.

1 ± 0.0%) (Fig. 4i). The SST+ cells also expressed high levels of HHEX protein, a transcription factor identified to be critical for δ-cell identity[46] as assessed via IF staining and we also verified the expression of GCG and PPY cell in the MAFB−/− cells (Fig. 4j; Supplementary Fig. 10b, c).

Finally, we assessed the expression of these hormones in the grafts of mice from Fig. 3a. We identified a strong correlation with our in vitro differentiated cells, whereby there was an overabundance of SST and PPY and less GCG producing cells in the MAFB−/− transplanted animals compared with MAFB+/+ transplanted cells (Supplementary Fig. 9g). These data highlight MAFB as a key transcription factor in lineage specification during human islet development.

**MAFB is essential for human β-cell identity.** We next investigated whether this phenotype could be rescued by re-expression of MAFB. To achieve this, we utilized a previously established doxycycline (DOX) inducible system in which iCas9 is replaced with a targeting cassette allowing for hygromycin selection (iHygro)[18]. This approach allows for precise reconstitution of one allele of MAFB within MAFB−/− cells, under the control of a reverse tetracycline-responsive element (M2rtTA) and a tetracycline-response element (TRE) (Supplementary Fig. 11a). Further details of the targeting and selection procedure to generate the DOX-inducible MAFB cell line (iHygroMAFB) are outlined in the Methods section.

DOX addition at the PP stage, the time point which coincides with *MAFB* gene expression under wild-type conditions (Fig. 1d), resulted in increased levels of MAFB protein at the β-like cell stage in MAFB−/− cells (Fig. 5a). FC analysis of the INS-GFP reporter revealed a significant increase in the levels of GFP (4 ± 2% vs. 12 ± 0.7%) at the β-like stage in DOX-treated cells compared with untreated cells (Fig. 5b, c). DOX addition to the parental MAFB−/− clone did not have any effect on differentiation. While C-PEP levels in the DOX-treated cells were significantly increased (13 ± 1% untreated vs. 19 ± 1.3% DOX-treated), SST levels were reduced (14 ± 7% untreated vs. 9 ± 4% DOX-treated) and both PDX1 and NKX6.1 levels were unchanged (47 ± 1% vs. 49 ± 14%) (Supplementary Fig. 11b–e). qPCR analysis revealed a similar pattern whereby *INS* transcript levels were significantly upregulated, *SST* and *PPY* transcript levels were significantly downregulated and levels of *GCG* and *PDX1* were unchanged. Interestingly the expression of *MAFA* was significantly upregulated in the rescue cells, indicating MAFB maybe required for induction of MAFA in human β-cells (Fig. 5d) and MAFB has previously been shown to bind and regulate MAFA transcription in a murine β-cell line[8]. Thus, re-expression of one copy of MAFB partially rescues the primary defects

observed in MAFB−/− cells, further supporting the MAFB-specific role.

Considering the reported high variability between hPSC lines in their capacity to differentiate[47], as well as variability within differentiation experiments, we sought to confirm the above findings in independent human β-cell models. First, utilizing commercially available induced PSCs (iPSCs), and the most efficient guide identified in the hESCs, we generated isogenic MAFB+/+, +/−, and −/− iPSCs (Supplementary Table 5). To account for variability between experiments, the control MAFB+/+ was set at 1 and experiments were normalized to their respective internal control for all FC datasets shown. In a similar manner to the hESCs, the iPSCs progressed through the DE and PP cell stages generating comparable efficiencies of SOX17 + /FOXA2 + and PDX1 + /NKX6.1+ cells (Fig. 5e–g). However, β-like cells (C-PEP + NKX6.1+ cells) were significantly reduced in the MAFB+/− and −/− cells compared with controls (1.0 vs. 0.5 ± 0.2 vs. 0.2 ± 0.1%) as assessed using both the FC and IF. Moreover, SST and PPY were upregulated compared with controls in both MAFB+/− and −/− cells, although PPY did not reach statistical significance (SST+ 1.0 vs. 3.2 ± 2.1 vs. 5.9 ± 1.3; PPY + 1.0 vs. 0.8 ± 0.6 vs. 2.1 ± 1.6) and C-PEP+ GCG+ cell numbers were significantly reduced (1.0 vs. 0.7 ± 0.2 vs. 0.2 ± 0.1). All FC data were confirmed using IF analysis for MAFB+/+, +/−, and −/− cells, and representative images are outlined in Fig. 5g and Supplementary Fig. 12a–e. This data demonstrate that re-organization of the endocrine cell compartment due to MAFB loss is conserved across independent hPSC lines.

Second, we used the EndoC-βH2 immortalized human β-cell line[48] to examine the effect of MAFB loss on established β-cell identity. This should eliminate any bias from using hPSCs to model β-cells. Using shRNA targeting MAFB and achieving ~50% knockdown efficiency as assessed via mRNA levels (Supplementary Fig. 13a), decreased the expression of *Pre-INS* transcript levels[49] was observed. Neither the β-cell transcription factor *NKX6.1* nor the endocrine marker *NEUROD1* were significantly altered. In contrast, *SST* was increased in MAFB knockdown cells and there were increases in *PDX1* and the δ-cell marker *HHEX* (Supplementary Fig. 13a). We also confirmed a similar pattern at the protein level whereby cells which expressed high levels of mCherry or GFP depicting lentiviral integration, had low levels of MAFB protein (Supplementary Fig. 13b) and had lower levels of INS and increased expression of SST as assessed via IF staining (Supplementary Fig. 13c). Notably, MAFB has previously shown to have an important role in GSIS of EndoC-βH1 cells in line with our observations[50]. These data highlight the requirement of MAFB in the generation and maintenance of human β-cell identity.

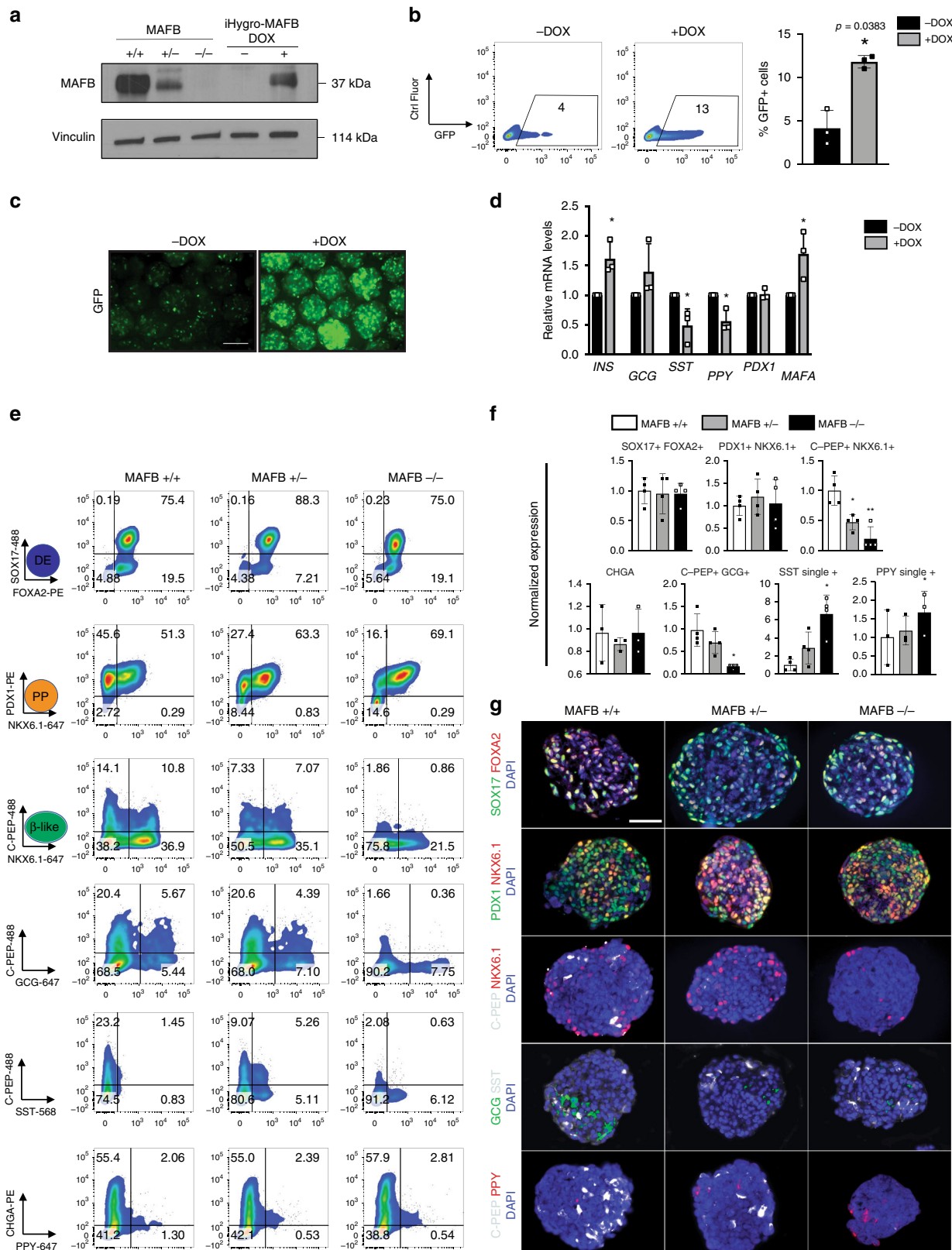

**MAFB elimination compromises α-cell differentiation**. Based on the above observations we aimed to further investigate the loss of GCG in human MAFB−/− cells given its reported role in α-cell development in mice. Using a modified version of a previously reported α-cell differentiation protocol[51], MAFB+/+ and −/− hESCs were differentiated in a stepwise manner toward glucose-responsive α-like cells as outlined in Fig. 6a. The precise differentiation conditions are outlined in Supplementary Tables 10 and 11.

In a WT INS-GFP background, GCG mono-hormonal cells are established by transitioning through a previously noted C-PEP+ GCG+ co-expressing stage[51] to a mono-hormonal GCG

**Fig. 5 MAFB is critical to β-cell identity. a** Representative Western blotting from three independent experiments for MAFB protein (37 kDa) expression in MAFB−/− and iHygroMAFB rescue cells with or without DOX as indicated. Vinculin (114 kDa) was used as a loading control. **b** Representative FC plots depicting the percentages of GFP+ cells and quantification at the β-cell stage ($n = 3$). P values by paired two-tailed $t$-test. **c** Representative GFP and Brightfield images from three independent experiments with or without DOX from live in vitro cell cultures at the β-cell stage. Scale bars, 100 μm. **d** The mRNA expression patterns of designated genes for islet hormones and selected transcription factors (*INS, GCG, SST, PPY, PDX1*, and *MAFA*) as measured by quantitative PCR analysis in the presence or absence of DOX ($n = 3$). P values by paired two-tailed $t$-test. *$P < 0.05$, **$P < 0.01$, ***$P < 0.001$, ****$P < 0.0001$. **e** Representative FC plots depicting the percentages of indicated markers at the indicated cell stages ($n = 3$ or 4, biologically independent samples for MAFB+/+, +/−, and −/− iPSC differentiation). **f** Quantification of the indicated cell populations from FC plots in (**e**), normalized to the unedited control, set to 1. **g** Representative IF images from three independent experiments at the DE, PP, and β-like cell stages of differentiation for the respective markers as indicated. Scale bars, 50 μm. P values by one-way ANOVA followed by Dunnett's multiple comparisons test were *$P < 0.05$, **$P < 0.01$, ***$P < 0.001$, ****$P < 0.0001$.

phenotype in the latter time-points. We confirmed that our strategy recapitulated previously reported stage specific criteria for generation of α-like cells via FC and IF analysis (Supplementary Fig. 14a–c). Differentiation of MAFB+/+ and −/− cells and FC analysis of intermediate stages highlighted a severe inability of MAFB−/− cells to generate GCG+ C-PEP+ cells compared with controls (D12; 14 ± 4% vs. 0.2 ± .02%) (D19; 32 ± 5% vs. 1 ± 0.3%) (D26; 17 ± 3% vs. 0 ± 0%) (D34; 10 ± 2% vs. 0 ± 0%) (Fig. 6b, c). Notably, the MAFB+/+ clone differentiated as expected in parallel to the parental INS-GFP cell line. Moreover, the total number of GCG+ cells produced was significantly downregulated at all time-points analyzed (D12; 22 ± 5% vs. 2 ± 0.6%), (D19; 47 ± 2% vs. 24 ± 2%), (D26; 51 ± 4% vs. 19 ± 0.5%), and (D34; 63 ± 3% vs. 28 ± 5.5%), likely due to an inability to generate the early stage bi-hormonal GCG+ C-PEP+ cells (Fig. 6d).

Analysis of MAFB+/+ and −/− cells at D26 indicated that in a similar manner to the β-like cell differentiations, expression levels of major transcription factors necessary for α-cells as measured via FC were not significantly altered between the two genotypes (ARX+ cells 64 ± 7% vs. 60 ± 3%, PAX6+ NKX2.2+ cells 61 ± 16% vs. 57 ± 6%). There was no significant difference in the levels of the β-cell transcription factors PDX1+ NKX6.1+ cells (11 ± 2.4% vs. 8 ± 6%) (Fig. 6e–g). In line with data from our hPSC to β-cell differentiation approach, qPCR analysis throughout the time course of α cell differentiation revealed reduced mRNA levels of *INS* and *GCG* in the MAFB−/− cells, while *SST* transcript levels were increased and *PPY* had a trend toward upregulation (Supplementary Fig. 14d). Notably, the levels of endocrine maker *CHGA* as well as the α- and β-cell promoting transcription factors *ARX* and *PDX1*, respectively, were unchanged (Fig. 6g).

IF analysis of D26 cells confirmed a reduction in the levels of INS and GCG in MAFB−/− cells, while ARX expression was similar to those of MAFB+/+ differentiations (Fig. 6h, Supplementary Fig. 15a, b). Moreover, SST was highly upregulated in MAFB−/− cells, in line with the mRNA data. We next assessed the in vitro glucagon secretion of MAFB+/+ and −/− cells in static hormone release assays and found MAFB−/− cells had significantly less total Glucagon content compared with controls (0.26 ± 0.08 (pg)/μg DNA vs. 0.06 ± 0.01 (pg)/μg DNA) (Fig. 6i). Interestingly, however, despite the lower levels of GCG in the MAFB−/− cells, they responded to glucose stimulation in a similar manner to MAFB+/+ cells, with reduced GCG secretion in response to high glucose as shown for individual experiments in Supplementary Fig. 15c.

## Discussion

The results of this study establish a requirement for MAFB in human β-cell formation and function. In the absence of MAFB, there is little or no impact on the early stages of pancreatic development, however, the cellular composition of the hormone-producing endocrine cell compartment is re-defined in favor of

somatostatin- and pancreatic polypeptide-producing cells, at the expense of insulin- and glucagon-producing cells. Utilizing differentiation strategies to generate human α- or β-cell counterparts and complementary results in three independent cell lines, we provide evidence that MAFB is a critical regulator of β-cell identity in humans. Furthermore, our study highlights use of hPSCs as a model for interrogating human pancreas development provides evidence for previously unappreciated roles of MAFB in endocrine cell lineage commitment.

Recent studies have recapitulated and expanded previous findings from GEMMs in terms of pancreatic development in humans. For example, the groups of Huangfu and Kieffer reported overlapping and non-distinct roles of PDX1, NGN3, RFX6, ARX, and PAX4 in early human and mouse pancreatic development and lineage specification[18,21,22]. Moreover, a small number of reports have taken advantage of hPSC models to study human disease associations which are poorly or only partially correlated in GEMMs[18–20]. Importantly, these reports elaborate on important limitations of using GEMMs such as patient specific mutations within transcription factors that are poorly correlated to human disease[3,52–54]. In MAFB KO mice, β-cell formation is merely delayed while α-cells are severely compromised, with no major changes in endocrine cell lineage specification[9,55]. Here we report a significant shift in the allocation of hormone-producing cells due to MAFB loss and highlight a requirement of MAFB in β-cell specification and function in humans.

Interestingly, there are no reported associations between MAFB and diabetes susceptibility. This is in contrast to other important transcription factors for pancreatic development including GATA6[56] or members of the hepatocyte nuclear factor (HNF) family[57–59] and glucokinase (GCK)[60], which are classified into maturity onset diabetes of the young (MODY) gene contributors of diabetes. Most recognized MODY-associated genes are implicated in the early stages of pancreatic development and endocrine cell number is often compromised upon removal of islet-enriched transcription factors such as NGN3[61] or NKX6.1[62]. However, in the pancreas MAFB appears to only have a role in the late stages of islet hormone-producing cell development and does not significantly affect the derivation of endocrine cell numbers. The relevance of this is not yet understood, although it has been demonstrated that MAFB KO mice exhibit neonatal lethality due to a plethora of developmental defects[63]. This may suggest essential roles for the transcription factor in other viable tissues such that human patients lacking MAFB would be unlikely to survive throughout development. In addition, MAFB has known roles in the cell specification of other organs including the brain, kidney, and immune system, demonstrating that it has a profound impact on diverse cell-fate decisions[39,63–65].

Intrinsic defects in β-cells are contributing factors in the dysfunction, de-differentiation and loss of β-cell mass observed in diabetes[66]. Several studies show that MAF factors directly bind to or control the expression of multiple TFs involved in insulin

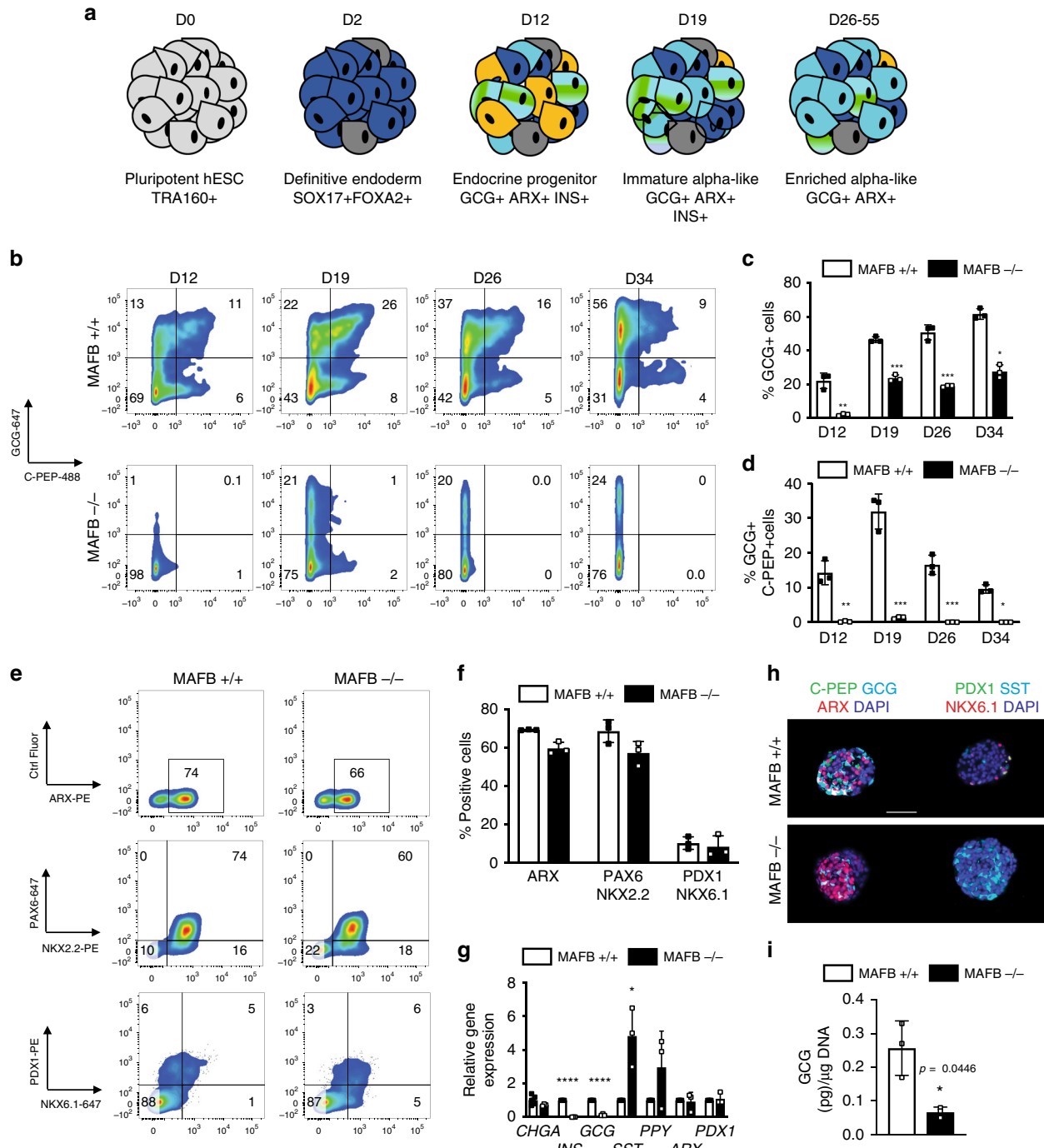

**Fig. 6 Loss of MAFB limits α-cell differentiation. a** Schematic outlining the differentiation protocol to generate α-like cells from hESCs. The key lineage markers are outlined at each stage. Chemicals and durations for each differentiation stage are indicated in the Methods section. d, day(s); hESC, undifferentiated hESC stage; DE, definitive endoderm stage (SOX17 + FOXA2+); EP, endocrine progenitor (CHGA + PDX1+); immature α-like cells (GCG + INS + ARX+) and enriched α-like cells (GCG + ARX+). **b** Representative FC plots at the indicated time point depicting the percentages of GCG + C-PEP+ cells and quantification at the DE stage. **c, d** Quantification of FC data outlined in (**b**) ($n = 3$). $P$ values by one-way ANOVA followed by Dunnett's multiple comparisons test were *$P < 0.05$, **$P < 0.01$, ***$P < 0.001$, ****$P < 0.0001$. Data are presented as individual biological replicates and represent the mean ± SD. **e** Representative FC plots at the α-like cell stage depicting the percentages of ARX+, PAX6+ NKX2.2+, and PDX1+ NKX6.1+ cells. **f** Quantification of FC data in (**e**) ($n = 3$, biologically independent samples). **g** The mRNA expression patterns of designated genes for islet hormones and selected transcription factors (*CHGA, INS, GCG, SST, PPY, ARX,* and *PDX1*) as measured by quantitative PCR analysis ($n = 3$, biologically independent samples). $P$ values by unpaired two-tailed $t$-test were *$P < 0.05$, **$P < 0.01$, ***$P < 0.001$, ****$P < 0.0001$. **h** Representative IF images from three independent experiments at the α-cell stage depicting GCG, INS and ARX and PDX1, SST and NKX6.1 in d26 α-cells from MAFB+/+ and −/− differentiations. Scale bars, 50 μm. **i** Quantification of total GCG content from MAFB+/+ and −/− cells at d26−34 in vitro ($n = 3$, biologically independent samples). $P$ values by paired two-tailed $t$-test.

signaling, including PDX1 as well the insulin gene itself[50,67,68]. Loss of the MAF factors appears at the top of a cascade of β-cell transcription factor loss in the process of β-cell de-differentiation[69], with results in rodents demonstrating that these factors are highly sensitive targets with respect to hyperglycemia or oxidative stress. The reported loss of MAFB expression in diabetic islets warrants further investigation results presented here indicate this has detrimental consequences as residual MAFB−/− β-like cells produced in our differentiations fail to generate functional glucose-responsive cells after transplantation and results in EndoC-βH2 cells support a critical role of MAFB in β-cell identity. While further investigations are necessary, our data provide evidence that MAFB promotes and maintains the β-cell transcriptional network, including MAFA and NKX6.1 for correct functioning of mature human β-cells. Prior data have demonstrated that MAFB regulates MAFA transcription[8] and in line with this, we find a reduction of MAFA in MAFB−/− cells which is reversed in MAFB rescue cells. Recent work has also demonstrated that MAFB does not compensate for MAFA loss in murine adult β-cells, providing additional evidence that these related factors have discreet and non-overlapping functions in β-cell biology[12], an important observation provided their differential expression in human vs. mouse adult islets.

In addition, our results raise questions about the precise role of MAFB in other pancreatic endocrine cell types. Most prominently, while MAFB has been identified as an important regulator of α-cell function based on results from MAFB GEMMs, there is evidence lacking for a role of MAFB in human α-cells, though it is expressed in these cells. Interestingly, in NGN3-Cre MAFB floxed KO mice the PPY producing cell was reportedly increased while INS and GCG producing cells were decreased and SST was unchanged[10]. Evidence for the role of MAFB in somatostatin producing δ-cells has not been reported, despite numerous next-generation sequencing studies identifying MAFB to be robustly expressed in human δ-cells. Instead, our data point to a potentially repressive role of MAFB whereby removal of MAFB allows for an abundance of SST+ NKX6.1− cells in the context of endocrine cell specification. Remarkably, MAFB was recently shown to repress SST expression in the developing brain and its loss resulted in an overabundance of SST+ cortical interneurons (CINs) perinatally[70], although the mechanism responsible for this remains elusive. This highlights the need to develop appropriate models of human δ-cell cultures, either by directed differentiation strategies or in a similar manner to the EndoC-βH2 β-cell line[48]. In addition, the prominence of GAST and PYY producing cells types in MAFB−/− cells warrants further study as their roles in human pancreatic biology are poorly understood[42,43].

Taken together, the current study in association with results from recent next-generation sequencing reports suggests that modeling human development in combination with gene editing approaches will promote our understanding of pancreas development in a human-specific context. These results complement studies performed in GEMMs and support the use of parallel approaches to define the particular role of transcription factors at defined developmental stages.

## Methods

**Maintenance of hESCs.** All experiments were performed using the NIH approved human embryonic stem cell (hESC) line MEL-1 (NIH approval number: NIH hESC-11-0139), in which a GFP protein was knocked into one allele of the endogenous insulin locus[25], designated INS^W/GFP hESCs. INS^W/GFP hESCs (Passages 35–58) were cultured on irradiated mouse embryonic fibroblasts (iMEFs, Thermo Fisher) feeder layers in hESC maintenance media comprised of DMEM supplemented with 4.16 ng/ml FGF-2 and 1x Glutamax. Confluent hESCs were dissociated into single-cell suspension by incubation with TrypLE Select (Gibco) and passaged every 3–4 days. Cells were maintained at 37 °C with 5% CO$_2$ and were routinely tested for mycoplasma contamination (Mycoprobe Detection Kit,

R&D Systems). hESC work was conducted according to NIH guidelines and approved by the Human Gamete, Embryo and Stem Cell Research (GESCR) Committee at UCSF, #11-05307. All chemicals and media components are outlined in Supplementary Tables 1, 2, 10, 11, and 12.

**Generation of clonal MAFB mutant lines.** INS^W/GFP inducible Cas9 (iCas9) hESCs were generated using the iCRISPR platform[26]. In brief, Neo-M2rtTA and Puro-Cas9 (Addgene 60843 and 58409) donor plasmids containing homology to the AAVS1 locus were site specifically integrated into the first intron of the constitutively expressed gene *PPP1R12C* at the *AAVS1* locus using a pair of TALENs (AAVS1-TALEN-L and AAVS1-TALEN-R; Addgene 59025 and 59026). This system allows for DOX-inducible Cas9 expression enabling efficient gene editing in the presence of gRNAs.

gRNAs targeting the MAFB coding region were designed using the online CRISPR design tool from Feng Zhang's laboratory (http://crispr.mit.edu/)[71] and three gRNAs with the highest predicted mutagenic efficiencies were used for generation of mutant lines (sequence information is outlined in Supplementary Table 3). gRNAs were synthesized via in vitro transcription (IVT) using the MEGAshortscript T7 kit (Life Technologies)[18].

Two days before gRNA transfection, INS^W/GFP iCas9 hESCs were treated with doxycycline (2 mg/ml). Cells were dissociated using TrypLE select and transfected in suspension with gRNAs or gRNA/ssDNA using Lipofectamine RNAiMAX (Thermo Fisher Scientific) according to manufacturer's instructions. Briefly 80 ng of the combined gRNAs and Lipofectamine RNAiMAX were mixed separately with Opti-MEM (Thermo Fisher Scientific) and incubated for 5 min at room temperature (RT). The two solutions were then combined together for 20 min at RT and subsequently added dropwise to 100,000 INS^W/GFP iCas9 hESCs in 24-well plates (Corning). Upon reaching confluency, cells were dissociated using TrypLE Select (Gibco) and re-seeded at low density (~1000 cells) onto iMEFs in 10-cm plates (Corning) in hESC Maintenance Media to allow for colony picking. The rest of the cells were subject to genomic DNA extraction. Genomic regions flanking the CRISPR target sites were PCR amplified using primers outlined in Supplementary Table 4 and then a T7 endonuclease I assay (T7EI) was performed according to manufacturer's instructions to determine the presence of genomic editing. Upon identification of successful editing, seeded cells were allowed to grow and form clonal colonies. Medium was changed every 2–3 days. Approximately 10 days later, individual colonies were manually picked, dissociated and replated into individual wells of 96-well plates pre-seeded with the iMEF feeder cells. Clonal lines were further expanded and stocked, genomic DNA isolated and a T7EI assay was performed on each individual clone to allow for identification of correctly targeted clones. Clonal cell lines carrying desired mutations were then subjected to TOPO blunt end cloning (Thermo Fisher Scientific) of PCR products to allow for segregation of heterozygous alleles and exclude potential contamination of cells with different genotypes. Isogenic WT clones which did not undergo genomic editing were also established as controls.

The Gibco^TM Episomal hiPSC Line was purchased from Thermo Fisher Scientific (#A18945). For generation of MAFB knockout hiPSCs. Cells were grown according to manufacturer's instructions to generate stock cell banks. For differentiation experiments, cells were adapted to cell culture conditions as described for the MEL-1 cells above using iMEFS. To generate isogenic MAFB+/+, +/−, and −/− clonal cell lines, Edit-R Modified Synthetic crRNA against MAFB along with tracrRNA were obtained from Dharmacon. The gRNA was resuspended to 160 μM in resuspension buffer (10 mM Tris HCL, 150 mM KCL pH 7.4) and stored at −80 °C until use. TracrRNA and gRNAs were incubated for 30 min at 37 °C to form a complex. Next, Cas9 RNP was added to the RNA complex for 15 min at 37 °C. iPSCs were dissociated into single cells using TrypLE Select (Gibco) and 100,000 cells were resuspended in P3 primary cell solution (Lonza). Cas9 RNP and the cell solution were then added to a Nucleocuvette (Lonza) and nucleofection was carried out using the setting CA137 on the Amaxa 4D-Nucleofector. Cells were replated in hESC Maintenance media on iMEFs (Thermo Scientific) in the presence of ROCK inhibitor for 24 h. Cells were allowed to recover until reaching confluency and then re-seeded at clonal density to allow for colony formation. Sequencing was carried out using the TOPO blunt end cloning (Thermo Fisher Scientific) strategy described above.

**Establishment of inducible gene expression lines.** For temporal control of MAFB cDNA activity in MAFB−/− hESCs, human MAFB cDNA (Roland Stein) was sub-cloned into the Hygro-iNGN3 plasmid (Addgene 75340) to generate Hygro-iMAFB donor plasmid. Two pre-designed gRNAs (AAVS1-cr1-ex and cr2-ex, Supplementary Table 13) which specifically target the iCas9 cassette used to generate the INS^W/GFP iCas9 hESCs[18] were obtained as Edit-R Modified Synthetic crRNA (Dharmacon) along with tracrRNA. Each gRNA was resuspended to 160 μM in resuspension buffer (10 mM Tris HCL, 150 mM KCL pH 7.4) and stored at −80 °C until use. TracrRNA and gRNAs were incubated for 30 min at 37 °C to form a complex. Next, Cas9 RNP was added to the RNA complex for 15 min at 37 °C. Clonal INS^W/GFP iCas9 hESCs were dissociated into single cells using TrypLE Select (Gibco) and 100,000 cells were resuspended in P3 primary cell solution (Lonza). Cas9 RNP, 5 μg/ml Hygro-iMAFB donor plasmid and the cell solution were then added to a Nucleocuvette (Lonza) and nucleofection was carried out using the setting CA137 on the Amaxa 4D-Nucleofector. Cells were replated in

hESC Maintenance media on DR4 iMEFs (Thermo Scientific) in the presence of ROCK inhibitor for 24 h (h). Cells were allowed to recover for 48 h and then subjected to Hygromycin (Invivogen) selection for 5 days. Cells were maintained on Hygromycin for three passages to ensure for efficient selection of targeted cells. Cells were then stocked and genomic DNA was extracted to verify correct insertion of the Hygro-iMAFB into the AAVS1 locus.

**hESC differentiation into α- and β-cells**. In β-cell differentiation assays, three independent mutant lines per MAFB+/− and −/− were analyzed in parallel with an isogenic WT control (generated from the same targeting experiments) as well as the parental cell line which underwent no genomic editing and experiments were repeated at least three independent times. Provided the consistency observed within the β-cell differentiation assays, one MAFB−/− clone along with one MAFB+/+ clone in parallel with the parental cell line were differentiated to α-cells. The precise differentiation protocols for α- and β-cell differentiations are outlined in Supplementary Tables 1, 2, 7, and 8, respectively. Briefly, hESC maintenance cell cultures were washed with PBS, dissociated to single cells with TryplE for 5–7 min at 37 °C and resuspended at a density of $5.5 \times 10^6$ cells per wells in suspension culture plates (Corning) using D0 media.

**Flow cytometry/fluorescence-activated cell sorting (FACS) analysis**. Spheres were collected and allowed to settle by gravity. Cells were washed once in PBS and dissociated by incubation in Accumax (Innovative Cell Technologies) at 37 °C with gentle agitation (5–7 min for DE samples, 8–10 min for PP samples and 12–15 min for β-like samples). For sorting, cell suspensions were filtered and resuspended in FACS buffer consisting of PBS (UCSF cell culture facility) containing 2 mM EDTA (Ambion) and 1% BSA (Sigma). Dead cells were excluded by DAPI (Sigma) staining. Cell sorting was performed on a FACSAria II (BD Bioscience). For flow-based analysis, dissociated cells were fixed with 4% paraformaldehyde (Electron Microscopy Science) for 15 min at room temperature, followed by two washes in PBS. Samples were either stored at 4 °C or immediately stained with antibodies as outlined in Supplementary Table 14. Data analysis was performed with FlowJo software (version 10.5.3). Mouse glucagon and mouse human C-peptide antibodies were conjugated in-house with Antibody Labeling Kits (Molecular Probes) according to manufacturer's instructions. Gating was based on stage matched samples that were unstained or negatively stained for the associated markers and an example of the strategy is shown in Supplementary Fig. 16e.

**Immunofluorescence analysis**. Spheres were fixed for 15 min at RT with 4% paraformaldehyde, followed by two washes in PBS and stored at 4 °C. Spheres were then embedded in 2% agar (Sigma), followed by dehydration, paraffin embedding, and sectioning at 5-μm thickness. For staining, sections were rehydrated and treated with and antigen retrieval solution (Biogenex) before blocking and incubation in primary antibodies overnight at 4 °C in blocking buffer (CAS-Block, Life Technologies with 0.2% Triton-X 100, Sigma). The next day, sections were washed three times in PBS with 0.1% Tween®20, Sigma and incubated with appropriate secondary antibodies for 45 min at RT in PBS-T. Alexa-conjugated secondary antibodies were purchased from Molecular Probes and used at 1:500 dilutions. Slides were washed in PBS-T and PBS before mounting in Vectashield. Nuclei were visualized with DAPI. Images were acquired using a Leica SP5 microscope or a Zeiss ApoTome. Primary antibodies used are outlined in Supplementary Table 15.

**Hormone content analysis**. Human C-peptide measurements were performed by assaying an aliquot of acidic ethanol lysed spheres or serum with commercially available ALPCO ELISA kits (human C-peptide: 80-CPTHU-CH01). Human Glucagon measurements were assayed using ALPCO ELISA kits (Glucagon ELISA: 48-GLUHU-E01). Briefly, samples were incubated for one hour in 2.8 mM glucose containing KRB to allow equilibration of cells. 2.8 mM buffer was removed and replaced with fresh KRB containing 2.8 mM glucose for 1 h followed by incubation for another hour in KRB containing 16.7 mM glucose. After the incubation period, buffers were collected and stored at −20 °C until use. Cells were lysed using Acid Ethanol to determine total hormone content (pg/ml) and total DNA from spheres was quantified by PicoGreen (Invitrogen) assay.

**Western blotting**. RIPA buffer (Pierce) supplemented with proteinase inhibitor cocktail (Roche) was used to extract total protein from cells and was measured using Bradford assay (Bio-Rad). Cell lysates were resolved on 10% or 4–20% acrylamide gradient SDS–PAGE gels (Bio-Rad). The samples were then electro-transferred to PVDF membranes (Bio-Rad) and immunoblotted with primary antibodies as outlined below. Immunoblotting with anti-GAPDH was used to confirm equal loading control. HRP-conjugated secondary antibodies (Jackson ImmunoResearch) were used and enhanced chemi-luminescence detection using Supersignal™ West Femto Maximum Sensitivity Substrate or SuperSignal™ West Pico Chemi-luminescent Substrate (Thermo Scientific) allowed for development of the signal. Primary antibodies are outlined in Supplementary Table 16. Uncropped and unprocessed scans of western blots are shown in Supplementary Fig. 16a–d.

**qPCR analysis**. Total RNA was isolated with either micro/mini RNeasy kit (Qiagen) and reverse-transcribed using the iScript cDNA Kit (Bio-Rad) according to manufacturer's instructions. SyberGreen qPCR analysis was performed on CFX Connect Real-Time System (Bio-Rad) using standard protocols. Primers were obtained from Integrated DNA Technologies (IDT) and sequence information is outlined in Supplementary Table 17.

**Transplantation studies**. NOD.Cg-Prkdcscid Il2rgtm1Wjl/SzJ (NSG) male and female mice at 8–12-weeks old were obtained from Jackson Laboratories. Mice were maintained at a constant humidity between 30 and 70% and temperature of 68–79°F, under a 12-h light/dark cycle and had free access to food and water until experiment initiation. For kidney capsule grafts, ~$5.0 \times 10^6$ differentiated cells in spheres were transplanted as described[72]. For glucose-induced insulin secretion, 5 weeks after transplantation, mice were fasted overnight and serum was collected before and after intraperitoneal administration of 2 g D-(+)-glucose/1 kg body weight (Sigma, G7021). Graft-bearing kidneys were removed for immuno-fluorescence analysis. Animals used in this study were maintained according to protocols approved by Laboratory Animal Resource Center, at UCSF.

**MAFB lentiviral knockdown in EndoC-βH2 cells**. Human EndoC-βH2 cells were grown in DMEM containing 5.6 mM glucose, 2% BSA, 50 μM 2-mercaptoethanol, 10 mM nicotinamide, 5.5 μg/mL transferrin, 6.7 ng/mL selenite, 100 units/mL penicillin, and 100 units/mL streptomycin[48]. For gene knockdown, cells were incubated with lentiviral particles carrying either shControl non-targeting sequences or sequences targeting MAFB (pLV[shRNA]-mCherry:T2A:Puro-U6>hMAFB[shRNA#1]; Vector Builder, Vector ID: VB180215-1122jqv) one day after plating. Cells were selected for efficient lentiviral infection by Puromycin treatment (2 μg/mL) for 72 h following incubation with the virus.

Cells were subject to IF staining with Rabbit α-MAFB, Novus (NB600-266), 1:500, and Goat α-SST, Santa Cruz (D-20) sc-7819, 1:500. RNA was collected using Trizol reagent (Life Technologies) 1 week following infection. The iScript cDNA synthesis kit (Bio-Rad Laboratories, Inc.) was used for cDNA synthesis. The quantitative (q)PCR reactions were performed on a LightCycler 480 II (Roche), and analyzed by the ΔΔCT method. Significance was calculated by comparing the ΔCT values.

**Single-cell RNA-seq and data processing**. ScRNA-Seq data was generated on the 10X platform (10X Genomics, Pleasanton, CA) according to Single Cell 3′ protocol (v2 Chemistry) recommended by the manufacturer[73]. The Cell Ranger software pipeline (version 2.1.1) [http://software.10xgenomics.com] was used to demultiplex cellular barcodes, map reads to the genome and transcriptome using the STAR aligner, and produce a matrix of gene counts versus cells. The R package Seurat (version 2.3.4)[33] was used to process the unique molecular identifier[74] count matrix, and to perform data normalization (gene expression measurements for each cell were normalized by total expression, and log-transformed), dimension-ality reduction, clustering, and differential expression analysis. DE analysis was performed with the R packages DESeq2 (v.1.16.0)[75] and MAST v.3.10[76].

**Chromatin immunoprecipitation, sequencing, and analysis**. EndoC-βH2 cells were cross-linked for 10 min with 1% formaldehyde, collected, and lysed in SDS buffer (Millipore Chromatin Immunoprecipitation Assay Kit. Cat#17-295). Soni-cated chromatin (Diagenode Bioruptor, 55.0 μg DNA) was incubated with anti-MAFB (Bethyl BL658) or rabbit IgG control (5 μg) in ChIP dilution buffer (Mil-lipore Chromatin Immunoprecipitation Assay Kit) at 4 °C overnight with rotation. Forty microliters of Dynabead slurry (Invitrogen 10004D Protein G Dynabeads) was added and rotated at 4 °C for 2 h. Beads were then washed, and DNA was eluted following manufacturer's instructions. Reverse cross-linked DNA was pur-ified using the Qiagen PCR purification kit. Library construction was performed by HudsonAlpha, using HiSeq v4 chemistry and single-ended sequencing was per-formed on an Illumina HiSeq2500.

The 50 bp reads from each sample were initially assessed FastQC 0.11.7[77]. Following quality-filtering and adapter trimming with TrimGalore 0.4.0[78], the remaining reads were aligned to the UCSC reference human genome (hg19, GRCh37) with Bowtie2 v. 2.3.5 and then further filtered prior to peak calling, which was performed with MACS2.1.0[79]. Regions that overlapped peaks from input DNA negative control sample were removed. Downstream interpretation was comprised of visualizing called peaks through the UCSC genome browser and IGV[80]. Last, one of the replicates did not yield good-quality peaks and was removed from our final analysis. All pertinent data were uploaded to ArrayExpress, accession E-MTAB-8612.

**Statistical analysis**. All statistical analysis was performed on Prism v. 7/8 for Mac (GraphPad Software). $P$ values and $n$ numbers are outlined in each individual figure. Microsoft Excel v.16.35 was used for data aggregation and storage.

**Reporting summary**. Further information on research design is available in the Nature Research Reporting Summary linked to this article.

## Data availability

The authors declare that all data supporting the findings Browser of this study are available within the article and its supplementary information files or from the corresponding author (M.H.) upon reasonable request. RNA sequences for the single-cell RNA-sequencing analyses reported in this paper have been deposited in the GEO database under the accession code GSE145347. MAFB ChIP-seq data from EndoC-βH2 cells was uploaded to ArrayExpress, under the accession code: E-MTAB-8612. MAFB ChIP-seq data from human islets are available at Islet Regulome[81] [http://www.isletregulome.org/] and can be viewed using the UCSC Genome browser[82] [https://genome.ucsc.edu/cgi-bin/hgTracks?db=hg19&lastVirtModeType=default&lastVirtModeExtraState=&virtModeType=default&virtMode=0&nonVirtPosition=&position=chr1%3A1%2D2&hgsid=815997045_p41DJpFchqeSccV95h3OjaUelETa]. The source data underlying Fig. 4a, b and Supplementary Figs. 5a–c, 6a–d; 7a and 8 are provided as a Source Data file.

## Code availability

The R package Seurat (version 2.3.4)[33] was used to process the unique molecular identifier count matrix[74], and to perform data normalization (gene expression measurements for each cell were normalized by total expression, and log-transformed), dimensionality reduction, clustering, and differential expression analysis. DE analysis was performed with the R packages DESeq2 (v.1.16.0)[75] and MAST[76].

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

## Acknowledgements

We thank members of the Hebrok laboratory and UCSF Diabetes Center for helpful comments and discussion. In particular, we thank Charity Juang for technical expertise with kidney capsule transplantations and Mei Lan Lei and Michael Saxton for excellent technical assistance. Flow cytometry experiments and image acquisition were supported by resources from the UCSF Flow Cytometry Core and Diabetes and Endocrinology Research Center (DERC) microscopy core P30 DK63720. Library preparation and QCs for RNA sequencing was conducted by Jim McGuire and Natasha Carli, PhD at the Gladstone Genomics Core. We thank Jean-Philippe Cartailler and the Vanderbilt Creative Data Solutions Shared Resource for performing ChIP-Seq data processing, analysis, and deposition. R.R. was supported by a Richard G. Klein Fellowship and JDRF Postdoctoral Fellowship (3-PDF-2018-588-A-N). Stem cell research in the laboratory of M.H. is supported by grants from the NIH (R01 DK105831, R01 DK090570). Research in the laboratory of R.S. is supported by (R01 DK090570).

## Author contributions

R.R., R.S., and M.H. conception and design of research; R.R., P.P.C., E.M.W., T.G.H., H.A.R., and J.S.L. performed experiments; R.R., P.P.C., E.M.W., R.S., and M.H. analyzed data and interpreted results; R.R. and S.G. performed bioinformatic analysis; R.R. and M.H. drafted the paper; R.R., P.P.C., T.G.H., E.M.W., H.A.R., J.S.L., S.G., R.S., and M.H. revised and approved the final version of the paper.

## Competing interests

M.H. is affiliated with Semma Therapeutics (Consultant and SAB member, Stock holder) and Encellin Inc. (SAB member, Stock holder). He holds stocks from Viacyte Inc. (Stock holder) and receives research support from Eli Lilly. H.A.R. is a consultant and SAB Islet member to Sigilon therapeutics and SAB member at Prellis Biologics. All other authors declare no competing interests.

## Additional information

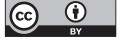

