## [Peer Review File · Nature Communications]

Reviewers' Comments:

Reviewer #1:

Remarks to the Author:

MAFB/MAFA are a classic example where mouse models may not fully recapitulate human biology, in that their expression patterns shows stage-specific differences. The authors use CRISPR to edit hPSC lines, and differentiate these to β -like cells, but also implement an alpha cell differentiation protocol. They demonstrate an essential stage-specific role of MAFB for endocrine cell type specification and maintenance of human β -cell identity, and confirm major findings in a human β -cell line.

The studies are generally very well executed, and provide a paradigm for how genome-edited pluripotent cells can be used to define regulators of beta cell differentiation. I have a number of points that need to be addressed (see below). A more general point is that it is inaccurate to convey the message that human in vitro cell models provide the final word for understanding human biology, as opposed to mouse models. In vitro human models are obviously extremely useful, as this study shows, but there are already several examples in which the results human in vitro models are incompatible with human genetics (as expected given that in vitro and in vivo differentiation do not necessarily proceed in the same environment and therefore do not have the same requirements). I would therefore suggest to slightly tone down some of the claims in this regard

Specific issues:

- MAFB mRNA is detected in the PP stage (Figure 1D), but not at the protein level. I would not emphasize the lack of phenotype in PP stages as if it was an unexpected finding.

- Figure 2E. Please separate channels, at least to enable visualization of major findings. Same for similar figures (4G, 5G). Some quantification is needed, and details regarding the number of differentiations and mice used.

- Figure 4 FACS:

Why are results so different in the representative examples in Figure 2 and 4? For example, C-peptide positive cells are >50% in WT in Fig 4, whereas in Figure 2 they are approx 30%
Why is the reduction of GCG+INS- cells relatively mild in this protocol, compared with the results shown in Figure 6. In fact Figure 4D shows an infinite increase in GCG+ only cells in the KO cells. Presumably this might be specific to the protocol but this needs some explanation.

- Figure 4, single cell genomics:

- Please provide UMAP representations in which WT and KO cells are clearly visible. Currently only KO cells are seen. This for ex does not allow seeing the reduced beta cells in the KO. Maybe using smaller dots, or side by side graphs.

- Can authors focus on DE in specific cellular subpopulations, rather than bulk. Why not use bulk RNA-Seq for this purpose? It would be much more quantitative.

- Figure S3 shows a different clustering than in main figures. Please explain why

Figure 5.

- Please provide further details on the exact timing of dox, and compare MAFB and GFP with WT cells.

- Does the result reflect increased number of INS+ cells or increased expression of INS on a per cell basis?

- If the cell lines have been created, can authors comment on whether MAFB overexpression improves maturation in WT cells?

- MAFA is probably not highly expressed in the beta like cells, but it is induced slightly in the MAFB re-expressed cells; Is MAFA reduced in the KO cells?

- Some of the effects observed here could be indirect, or specific to this in vitro differentiation model. What is the relationship between MAFB-regulated genes in this system and MAFB-bound genes in human islets?
- All rescue experiments should show comparisons with wild type cells instead of just KO cells
- Fig 5F. Throughout the manuscript control points should show variation in controls, rather than pegging all values to 1 (its of course ok to normalize to 1 median/average). Statistical analysis in 5F should probably be a paired non parametric test or other tests appropriate for this data distribution. This is applicable to other similar experiments.
- Figure S1 D7 and D9 look remarkably identical, please check that they have not been duplicated
- Figure 6H. GCG seems unaltered and ARX seems to increase, does NKX6.1 increase? This is probably because the channels need to be separated, or because it is not a representative example.

Other points:

- Figure S2. Why are authors only showing P value < 0.05 in red, when in beta like cells the threshold is 0.1, and grey cells are > 0.1. Similar concepts apply elsewhere. Please use consistent adj p value cutoffs throughout the manuscript.
- Fig 1C could benefit from labelling stages
- The text refers to "genomic sequencing", probably best to say targeted sequencing of the genome-edited locus to avoid confusion
- GATA6 is mutated in neonatal diabetes but is usually not classified as MODY
- Use dots for experimental averages in the bar plots, e.g. Fig. 2A-D, and specific number of independent experiments in the legends
- Figure 6H should show separate images to support conclusions reported in the text.
- MAFB antibody should be described in Table S12
Which negative controls are used for FC? (unstained cells, control antibodies?) Which fluorophores are used? This is not always described.
- There were several typos, these are just examples:
"In line with this, MAFA expression increases in an age-dependent manner in both human and mouse b-cells, and is not robustly *expressed* in human beta cells until after 10 years of age"
"These results indicate that MAFB loss alone is sufficient to compromise the composition of the endocrine cell composition/compartement"

Reviewer #2:

Remarks to the Author:

This a carefully performed study with many quantitative data that aims at studying the function of the transcription factor MAFB in the human endocrine pancreas. The study uses loss-of-function approach in a model of hPCS towards pancreatic endocrine cell development. Data indicate that MAFB is an important regulator of cell fate decisions in the human pancreas. Below is a number of suggestions that aim at further improving the quality of the manuscript. I believe that the authors may have all the data to answer the suggestions.

General comments

- Primary human pancreatic beta cells express both MAFA and MAFB. I understand, may be wrongly, that hPSCs derived beta cells express very limited levels of MAFA, which is also the case for EndoC-BH2 used in one experiment. Could the authors discuss the low MAFA expression level (if the case) as a potential limitation of their work?
 - Page 5: The sentence "Due to pronounced growth of the graft, likely due to the higher proliferative capacity of the PP cells (discussed below) ..." The sentence is cryptic and I do not see how it is discussed "below"
 - Page 5: Is it expected, based on the literature, that MAFB^{+/+} PP cells do not give rise to Hormone⁺ cells following 8 weeks of transplantation?
 - Page 5: "human C-peptide levels (29±5.2 pg/ml vs. 157±37 pg/ml)". Would be interesting to compare such quantitative data to the ones obtained by other groups.
 - Page 5: What about insulin content per beta cells derived from PSCs, compared to primary human beta cells?
 - What remains a bit unclear for me is whether in the absence of MAFB, there is a decrease in the number of beta cells, or in the expression of a set of beta cells-specific genes. This may be discussed.
- Page 7, second chapter: "Similarly, we did not detect any difference in the levels of NGN3 protein levels between MAFB ^{+/+} and ^{-/-} cells...". Where is it shown?
- Page 7, second paragraph: "We also investigated whether there was a difference in the rate of proliferation". I understand that the authors performed Ki67 staining without looking at specific cell types. Did the authors perform some types of double staining?
 - Fig. 5: It would be really interesting to have MAFB^{+/+} cells on the same WB for comparison.
 - Page 8: "Interestingly the expression of MAFA was significantly up-regulated in the rescue cells". What are the Ct levels of MAFA (and MAFB) under the different conditions (MAFB^{+/+}; MAFB^{-/-}; MAFB^{-/-} + rescue).
 - On Fig. 2, there is a 10-fold decrease in the number of C-Pept⁺ in MAFB^{-/-} cells. Based on Fig. 5, there is a qualitative trend to revert this phenotype upon MAFB reexpression, but it did not reach values obtained in control conditions. The authors may discuss this point.
 - I am a bit puzzled by the biological relevance of the chapter "MAFB elimination compromises a-cell differentiation". I read that in the in vitro model, "GCG mono-hormonal cells are achieved by transitioning from a previously noted C-PEP⁺ GCG⁺ co-expressing stage". I thought that in vivo (at least in the mouse), it is not the case. The difference between mouse and human or the limit of the in vitro system should be highlighted.
 - Fig. 6G. CHGA levels are not changing, while INS and GCG mRNA are strongly decreased by more than 10 folds. I have the feeling that the increase in SST and PPY does not compensate the decrease in INS and GCG. Is there CHGA⁺ cells negative for the above endocrine markers?
 - Fig. 6I: would be nice to have the data as absolute and not relative values.

Minor comments

- Introduction: is MAFB expressed in PP cells?

- The right panel on Fig. 1C with MAFB is important but a bit difficult to read. It would be nice to deconstruct the panel with each individual marker.
- What about the long-term survival of EndoC-BH2 cells following MAFB knock-down? Do the cells continue growing?
- Why did the authors measure pre-INS transcript in EndoC-BH2 and not INS transcripts?
- Page 7 "or the b-like cell stage ($4.6 \pm 1.6\%$ vs. $5.7 \pm 2.4\%$) (Figure S4E,F)." It may be S5 instead of S4

Raphael Scharfmann

Reviewer #3:

Remarks to the Author:

In this manuscript, Russell et al. describe the role of transcription factor MAFB in differentiation of human pancreatic beta- and alpha-cells using hPSCs, iPSCs and EndoC-betaH2 cells, and demonstrate that MAFB regulates human endocrine cell lineage commitment. The loss of MAFB resulted in increased population of somatostatin (SST)- and pancreatic polypeptide (PPY)-expressing cells at the expense of glucagon (GCG)- and insulin (INS)-expressing cells. Induced expression of MAFB in MAFB^{-/-} cells partially rescued the phenotype above. Altered expression of transcription factors NKX6.1 and HHEX in addition to hormones was also demonstrated.

Results are clearly demonstrated with careful examination and generally convincing. However, many experiments are not described in sufficient detail and several important control data are missing. Additionally, results described in this manuscript are generally in line with published findings elucidated by analyses of genetically engineered mouse models (GEMMs) and are expected, although authors emphasize that their results are considerably different from those from GEMMs.

Major comments:

- 1) Fig. S1A-E shows only two results of MAFB^{+/+}, which is important as a control. Therefore, the experiments with MAFB^{+/+} hESC (and iPSCs) should be performed at least in triplicate as the minimum number.
- 2) Please clarify how many mice and sections were examined for each immunofluorescence (IF) data, Figs. 1C, 1H, 2E, 3C, 4G, 4J, 5C, 5G and 6H, and show statistical analysis of these results. In addition, what kind of PPY antibody was used for IF? Is it specific to PPY?
- 3) Results from GEMMs revealed that knockout of MAFB has not minimal but certain effect on murine beta- and alpha-cell development as described in references 7 and 8, which shows significantly reduced number of beta- and alpha-cells, while total number of endocrine cells was unchanged. It has also been described that number of somatostatin-expressing cells and ghrerin-expressing cells is unaffected in MAFB KO mice (Nishimura et al., Dev Biol 314, 443-456, 2008). Authors should discuss these previously reported results.
- 4) Fig. 6I clearly shows that loss of MAFB resulted in reduced glucagon content and that glucose-regulated GCG secretion was similar between MAFB^{+/+} and MAFB^{-/-} cells. How are the INS content and secretion of MAFB^{+/+} and MAFB^{-/-} beta-like cells in vitro? Additionally, can SST induced in MAFB^{-/-} cells be secreted from cells at physiologically meaningful level?
- 5) Regarding gene expression data of Fig. S2D and S2E (PP stage) and Fig. S4B (beta-like cell

stage), please show expression of HHEX, MAFA, PAX4, ARX, PAX6, CHGA, and ALDH1a3, the dedifferentiation marker. To mention that "without affecting total numbers of endocrine cells as marked by CHGA expression" in page 7, line 8-9, please show evidence.

Minor points:

6) In Fig. 3B, please show blood glucose data. How many times were islet transplantation experiments repeated?

7) Page 7, line6: "Figure 5C" should be "Figure 4C". Page 7, line 8: "to compromise the composition of the endocrine cell composition" should be "to compromise the composition of the endocrine cells".

8) Fig. 5A-D: It seems that number of INS-GFP cells in MAFB^{-/-} cells with DOX-induced MAFB expression does not reach the level of those in MAFB^{+/+} cells. In Fig. 5A please show quantitative data of MAFB expression levels in MAFB^{+/+} cells together with MAFB^{-/-}, iHYGRO-MAFB-DOX and iHYGRO-MAFB+DOX cells.

We would like to thank the reviewers for their fair and constructive review of our manuscript. We agree with their comments and have addressed them in full wherever possible as outlined in detail below.

Reviewers' comments:

Reviewer #1 (Remarks to the Author):

MAFB/MAFA are a classic example where mouse models may not fully recapitulate human biology, in that their expression patterns shows stage-specific differences. The authors use CRISPR to edit hPSC lines, and differentiate these to β -like cells, but also implement an alpha cell differentiation protocol. They demonstrate an essential stage-specific role of MAFB for endocrine cell type specification and maintenance of human β -cell identity, and confirm major findings in a human β -cell line.

The studies are generally very well executed, and provide a paradigm for how genome-edited pluripotent cells can be used to define regulators of beta cell differentiation. I have a number of points that need to be addressed (see below). A more general point is that it is inaccurate to convey the message that human in vitro cell models provide the final word for understanding human biology, as opposed to mouse models. In vitro human models are obviously extremely useful, as this study shows, but there are already several examples in which the results human in vitro models are incompatible with human genetics (as expected given that in vitro and in vivo differentiation do not necessarily proceed in the same environment and therefore do not have the same requirements). I would therefore suggest to slightly tone down some of the claims in this regard.

We thank the Reviewer for the appreciation of our work and the constructive criticism. We do not want to convey a general message that human in vitro models make the analysis of mouse models obsolete, just that they can be useful to study in parallel approaches to highlight particular species-specific differences. We have followed the suggestion by the Reviewer and toned down the language in the manuscript regarding this issue.

Specific issues:

- MAFB mRNA is detected in the PP stage (Figure 1D), but not at the protein level. I would not emphasize the lack of phenotype in PP stages as if it was an unexpected finding.

We thank the reviewer for this suggestion and as we completely agree, we have updated the text to reflect this.

- Figure 2E. Please separate channels, at least to enable visualization of major findings. Same for similar figures (4G, 5G). Some quantification is needed, and details regarding the number of differentiations and mice used.

Channels have now been separated.

The FACS quantification in **figures 4 and 5** represents quantitative data from matched experiments and representative IF images are shown for reference from at least 3 independent experiments.

Transplantation experiments were performed one time in three independent NSG mice per group. There were no outliers and the data are consistent with the *in vitro* experiments. The figure legend has now been updated to reflect this.

We have generated supplemental figures demonstrating single channels for composite images where requested by all reviewers throughout the manuscript.

- Figure 4 FACS:

Why are results so different in the representative examples in Figure 2 and 4? For example, C-peptide positive cells are >50% in WT in Fig 4, whereas in Figure 2 they are approx 30%

This is an inherent flaw with stem cell differentiations that we recognize. It is also recognized within the field of stem cell differentiations in general that not all differentiation experiments proceed with the same efficiency.

Examples include:

See Figure 1F - (Wesolowska-Andersen et al., 2020)

See Figure 2C - (Velazco-Cruz et al., 2019)

Despite the fact that our stem cell differentiation protocol is reliable and provides reproducible results, we do acknowledge that batch to batch effects have a measurable influence. We stand by the overall message that in the loss of MAFB, the relative compositions of endocrine cell types are significantly altered.

In an ideal scenario, we would have 5-6 'perfect' experiments in which all assays are performed but generating this quantity of material and processing it is not feasible. We do quality control of differentiations by looking for high expression of markers at the various differentiation stages (eg. SOX17 and FOXA2 at DE or PDX1 and NKX6.1 at PP stage) and believe the data is representative of the observed phenotype and we precisely used this raw representation of data in the case of the INS-GFP cells to reflect this.

In the case of the iPSCs, we used a normalization step where the control is normalized to 1 for simplicity (**Figure 5F**). Here we again see that the relative number of C-pep positive cells is significantly reduced in independent (internally controlled) experiments for MAFB loss of function cells.

Why is the reduction of GCG+INS- cells relatively mild in this protocol, compared with the results shown in Figure 6. In fact Figure 4D shows an infinite increase in GCG+ only cells in the KO cells. Presumably this might be specific to the protocol but this needs some explanation.

We agree that this is an interesting point, although data from other groups demonstrates the presence of GCG+INS+ cells that likely serve as alpha cell progenitors. Through our beta-cell differentiation protocol, we are promoting the induction of NKX6.1 and INS and as a result, we also unintentionally appear to generate other hormone producing cell types such as Glucagon producing cells. This has previously been noted in the field and remains an active area of research as to how one can prevent or further differentiate cells past this poly-hormonal positive stage (Velazco-Cruz et al., 2019) (Rezania et al., 2014) (Rezania et al., 2011). Notably, this polyhormonal cell state has previously been recognized in the human fetal pancreas (De Krijger et al., 1992).

In the MAFB +/- WT context, the majority of GCG+ cells appear to co-express C-Peptide. However, all our data reflect an overall loss of GCG in loss of MAFB conditions, with the remaining cells being single hormone. This finding is highlighted dramatically in **Figure 6** in which we are using conditions to predominately generate alpha cells over other islet cell types.

- Figure 4, single cell genomics:

- Please provide UMAP representations in which WT and KO cells are clearly visible. Currently only KO cells are seen. This for ex does not allow seeing the reduced beta cells in the KO. Maybe using smaller dots, or side by side graphs.

Unfortunately, as there are more KO cells, they tend to be more prominent in the plot. We have played around with many versions and present the best updated UMAP representation as requested. The individual UMAPs are for MAFB +/- and -/- are shown in **Figure S6B**.

- Can authors focus on DE in specific cellular subpopulations, rather than bulk. Why not use bulk RNA-Seq for this purpose? It would be much more quantitative.

We completely agree that using DE analysis on individual subpopulations is of great interest. We now provide DE analysis of the individual sub-populations in **Table S7,8,9** for reference.

However, we tried to highlight the primary observations clearly and are interrogating this data further as the basis for eventually proceeding with additional future studies.

- Figure S3 shows a different clustering than in main figures. Please explain why

Figure S3 has now become **Figure S6** in the revised version of the manuscript. The differences pointed out by the reviewer relate to the Individual sample clustering for MAFB +/- and -/- cells in the supplemental data (**Figure S6**), whereas in the main text, we provide an overview of the merged UMAPs to allow for cross-comparison of the samples and relative transcripts.

Figure 5.

- Please provide further details on the exact timing of dox, and compare MAFB and GFP with WT cells.

DOX was added at D9 at the onset of PP specification to allow for accumulation of protein.

Wt cells and rescue cells are now shown on the same western blot in **Figure 5A**. The text has now been updated to include this information.

- Does the result reflect increased number of INS+ cells or increased expression of INS on a per cell basis?

We measured the mean fluorescence intensity (MFI) of the GFP. Although there is a trend towards higher MFI in the MAFB-/- cells, thus indicating an increased expression of INS per cell, this does not reach statistical significance (**Figure S11F,G**). Therefore, we conclude that the result primarily reflects an increase in the number of INS expressing cells in line with our previous observations.

- If the cell lines have been created, can authors comment on whether MAFB overexpression improves maturation in WT cells?

We agree that this is an interesting point and we attempted to generate these cells. However, in the absence of DOX, these cell lines did not differentiate efficiently compared to the unedited control. We believe this maybe due to sub-cloning and plan to further investigate this in the future as well as in the context of MAFA overexpression. Considering the substantial amount of work that needs to be done, we consider both approaches are outside the scope of this manuscript.

- MAFA is probably not highly expressed in the beta like cells, but it is induced slightly in the MAFB re-expressed cells; Is MAFA reduced in the KO cells?

Yes, but only marginally and it does not reach significance. MAFA qPCR results are now shown in Figure S7C.

- Some of the effects observed here could be indirect, or specific to this in vitro differentiation model. What is the relationship between MAFB-regulated genes in this system and MAFB-bound genes in human islets?

This is an interesting point and it is certainly of interest to determine how MAFB regulated genes in our hPSC system correspond to MAFB bound genes in primary human islets.

To address this, we mined previously published data sets (Pasquali et al., 2014) in which they perform MAFB ChIP seq in two independent human islets preps. Notably, they use the same antibody (Sigma Prestige) that we used in the current study as well as mature human islets from adults. Interestingly, when we look at the tracks for MAFB binding peaks in the top 5 up- and down-regulated genes in our *in vitro* system, we find a strong correlation of genes with peaks near the promoter region that are downregulated upon MAFB loss (eg. *INS*, *TTR*, *CRYBA2*, *ACVR1C*). These data are now shown in Figure S8 of the updated manuscript.

Additionally, we performed MAFB ChIP seq in the EndoC-BH2 cells using an alternative MAFB antibody (Bethyl Laboratories). The objective here was to determine beta-cell specific MAFB binding peaks and notably, we identify a very strong overlap of genes bound in human islets that are deregulated in our hPSC upon loss of MAFB. Noticeably, MAFB binds to the promoter region of *INS*, *TTR* and *CRYBA2*, genes that have established roles in alpha and beta cell function. Together, we believe these data together provide strong evidence that the effects observed in our hPSCs demonstrate a specific role of MAFB in the human islet.

- All rescue experiments should show comparisons with wild type cells instead of just KO cells

We now include a western blot (Figure 5A) showing expression of MAFB $+/+$, $+/-$ and $-/-$ cells on the same blot as iHYGRO-MAFB - or + DOX treated cells. The levels of MAFB in the rescue cell line are lower than in the wildtype cells and more aligned with the MAFB $+/-$ cells. Notably, we replace MAFB in only one allele of the *AAVS1* locus as described in the materials and methods. These experiments were done in parallel with experiments shown in Figure 2C as reference.

- Fig 5F. Throughout the manuscript control points should show variation in controls, rather than pegging all values to 1 (its of course ok to normalize to 1 median/average). Statistical analysis in 5F should probably be a paired non parametric test or other tests appropriate for this data distribution. This is applicable to other similar experiments.

We thank the reviewer for this important request and have now updated the figures throughout the manuscript to show variation in controls.

Statistics have also been updated throughout using paired testing when appropriate. Please see individual figure legends for details.

- Figure S1 D7 and D9 look remarkably identical, please check that they have not been duplicated
We thank the reviewer for catching this.

Due to a copy/pasting error and the high degree of similarity between the FACS plots, the plot in **Figure S1 (now Figure S2)** for D7 had been duplicated in D9. The correct FACS plots are now shown in the revised version of the manuscript.

- Figure 6H. GCG seems unaltered and ARX seems to increase, does NKX6.1 increase? This is probably because the channels need to be separated, or because it is not a representative example.

The channels have now been separated which we believe aids with data interpretation.

Flow cytometry quantification shown in **Figure 6E,F** from matched experiments support our IF data from 3 independent experiments.

Other points:

- Figure S2. Why are authors only showing P value < 0.05 in red, when in beta like cells the threshold is 0.1, and grey cells are > 0.1. Similar concepts apply elsewhere. Please use consistent adj p value cutoffs throughout the manuscript.

We apologize for this oversight. The same thresholds of $pval > 0.1$ (in grey) and $logFC > 0.5$ and $pval < 0.1$ (in red) were used throughout for the single cell sequencing analysis.

- Fig 1C could benefit from labelling stages

We have now labelled the different stages as suggested.

- The text refers to “genomic sequencing”, probably best to say targeted sequencing of the genome-edited locus to avoid confusion

We appreciate the suggestion and have now updated the text to clarify this point.

- GATA6 is mutated in neonatal diabetes but is usually not classified as MODY

We had not intentionally classified GATA6 as a MODY gene but one that has disease associations and developmental defects when mutated in the pancreas. We thank the reviewer for pointing this out and have now updated the text to clarify this point.

- Use dots for experimental averages in the bar plots, e.g. Fig. 2A-D, and specific number of independent experiments in the legends

The graphics have now been updated and the number of independent experiments (n) is clarified in the legends throughout the text.

- Figure 6H should show separate images to support conclusions reported in the text.

We now include separate images in the supplemental data to support our conclusions (**Figure S15**).

- MAFB antibody should be described in Table S12

Which negative controls are used for FC? (unstained cells, control antibodies?) Which fluorophores are used? This is not always described.

Unstained aliquots of samples were used as negative controls for setting gates.

Fluorophores are now indicated in the materials and methods in new **Table S14**.

- There were several typos, these are just examples:

“In line with this, MAFA expression increases in an age-dependent manner in both human and mouse b-

cells, and is not robustly *expressed* in human beta cells until after 10 years of age”
“These results indicate that MAFB loss alone is sufficient to compromise the composition of the endocrine cell composition/compartments”

We apologize for these oversights. The text has now been updated to correct for these errors.

--

Reviewer #2 (Remarks to the Author):

This is a carefully performed study with many quantitative data that aims at studying the function of the transcription factor MAFB in the human endocrine pancreas. The study uses a loss-of-function approach in a model of hPSC towards pancreatic endocrine cell development. Data indicate that MAFB is an important regulator of cell fate decisions in the human pancreas. Below is a number of suggestions that aim at further improving the quality of the manuscript. I believe that the authors may have all the data to answer the suggestions.

We thank the Reviewer for constructive feedback and well documented suggestions/queries that help us to clarify and strengthen the take-home messages of the current study. We have followed the suggestions of the Reviewer and believe that for the most part, have been able to incorporate these points into the manuscript.

General comments

- Primary human pancreatic beta cells express both MAFA and MAFB. I understand, may be wrongly, that hPSCs derived beta cells express very limited levels of MAFA, which is also the case for EndoC-BH2 used in one experiment. Could the authors discuss the low MAFA expression level (if the case) as a potential limitation of their work?

Yes, it is correct that hPSC derived beta cells and also EndoC-BH2 express very low levels of MAFA. This is recognized as a limiting factor in our current study.

However, recent work has established that MAFA is not robustly expressed in human islets until after ~10 years of age (Arda et al., 2016) (Cyphert et al., 2018), while its expression increases in rodents prenatally with peak expression soon after birth (Hang et al., 2014) (Cyphert et al., 2018). Additionally, the latter study demonstrated that MAFA and MAFB have alternative functions that are not directly compensatory. Thus, while MAFA expression appears to rescue MAFB loss in the context of murine beta cells, this does not appear to be the case in humans as beta cells are specified before 10 years of age. Moreover, transplantation of our MAFB^{-/-} cells into NSG animals (**Figure 3**) demonstrates that in the absence of MAFB, there is an inability to generate functional beta-cells in this system.

- Page 5: The sentence “Due to pronounced growth of the graft, likely due to the higher proliferative capacity of the PP cells (discussed below) ...” The sentence is cryptic and I do not see how it is discussed “below”

We apologize for not stating this point more clearly. We are referring to revised **Figure S9E**, where we measured the proliferation rate of PP cells and beta-like cells using KI67. The PPs have a higher number of proliferating cells *in vitro* and in our mice that received the PP cell transplant, this graft grew larger than the graft that received the more differentiated beta-like cells. We have updated this paragraph in the text to avoid confusion.

- Page 5: Is it expected, based on the literature, that MAFB^{+/+} PP cells do not give rise to Hormone⁺ cells following 8 weeks of transplantation?

It is difficult to reconcile differences between different differentiation protocols, transplantation sites, cell numbers and maturation status, which is usually based on characterization of just two factors such as PDX1 and NKX6.1 to define a cell population such as a pancreatic progenitor. However, the report by Bruin et al., 2013 used a similar approach whereby S4 pancreatic progenitor cells were transplanted under the kidney capsule of diabetic SCID-beige mice and there were no functional hormone producing cells observed until 16 weeks post transplantation (Bruin et al., 2013).

- Page 5: “human C-peptide levels (29±5.2 pg/ml vs. 157±37 pg/ml)”. Would be interesting to compare such quantitative data to the ones obtained by other groups.

- Page 5: What about insulin content per beta cells derived from PSCs, compared to primary human beta cells?

As mentioned in the previous comment, direct comparisons are difficult.

In this precise case, we did not set out to assess human C-peptide levels in relation to a standard such as the Islet equivalent (IEQ) model (Ricordi et al., 1990). Our group has previously reported human C-peptide levels from stem cell derived beta-like cells in relation to IEQs which is more amenable to comparison of stem cell derived secretion by other groups (Russ et al., 2015) (Nair et al., 2019).

We fully acknowledge that our cells at this stage (d20) are immature or resemble more like juvenile beta cells and do not possess the full functional capacity of human beta cells (Nair et al., 2019). Comparisons in this experimental set-up were not the primary focus and we refer to our recently published data that addressed this point. Additionally, the majority of results in this study derive from using the INS-GFP cell line in which one copy of the INS allele is replaced by GFP.

Taken together, we do not believe the cells we are producing in the context of this manuscript, eg. immature d20 beta cells, are appropriate to draw conclusions for their equivalence to human islets and we recognize this as a potential limiting factor of this study.

- What remains a bit unclear for me is whether in the absence of MAFB, there is a decrease in the number of beta cells, or in the expression of a set of beta cells-specific genes. This may be discussed.

Our data reveal a significant decrease in the number of beta cells due to loss of MAFB (**Figure 2C,D**) which can be partially rescued by re-introducing MAFB (**Figure 5A-C**). Our RNA sequencing data indicate that INS transcript is one of the most differentially expressed genes between MAFB ^{+/+} and ^{-/-} cells and ChIP-seq data (**Figure S8**) reveals that MAFB binds to the *INS* promoter in human islets and EndoC-BH2 cells. We believe the primary reason for a reduced number of beta cells is an inability to activate *Insulin* gene transcription.

Page 7, second chapter: “Similarly, we did not detect any difference in the levels of NGN3 protein levels between MAFB ^{+/+} and ^{-/-} cells...”. Where is it shown?

We apologize for this oversight. We now clarify in the text that there was no significant difference in NGN3 mRNA levels as outlined in **Figure S5E** and **Figure S7B**.

- Page 7, second paragraph: “We also investigated whether there was a difference in the rate of proliferation”. I understand that the authors performed Ki67 staining without looking at specific cell types. Did the authors perform some types of double staining?
 To date, we have not performed Ki67 staining on specific cell types, although we agree that this will be of interest in follow-up studies.

- Fig. 5: It would be really interesting to have MAFB+/+ cells on the same WB for comparison. The western blot has now been updated to show MAFB +/+, +/- and -/- cells on the same blot as the rescue conditions (Figure 5A).

- Page 8: “Interestingly the expression of MAFA was significantly up-regulated in the rescue cells”. What are the Ct levels of MAFA (and MAFB) under the different conditions (MAFB+/+; MAFB-/-; MAFB-/- + rescue).

	Sample No.	Ct	Sample
MAFA	B01	31.23	MAFB -/-
	B02	30.38	MAFB +/-
	C02		No template control - H2O
MAFB	E01	23.14	MAFB -/-
	E02	23.39	MAFB +/-
	F02		No template control - H2O

Response to reviewers - Figure 1. qPCR results for MAFA and MAFB.

The left panel shows melting curves of qPCR for MAFA and MAFB as indicated by arrows. The right panel indicates an example of Ct values for MAFA or MAFB as indicated MAFB+/+ and -/- samples for either. Also included is a no template control (H2O) which shows no melting curve or amplification using either primer set. The bottom panel is a table showing representative Ct values for MAFB+/+ and -/- cells for either MAFA or MAFB qPCR.

Overall, there is a trend of downregulation in MAFB-/- cells as shown in updated Figure S7B, although this does not reach statistical significance.

We do not detect significant changes in MAFB at the mRNA level across the different genotypes, likely due to the nature of the frameshift mutations introduced. Depending on the individual experiment, we

observe a range of Ct values for the MAFB^{-/-} cells but these in line with the values obtained for the MAFB^{+/+} and +/- counterparts with an example shown above for reference.

		MAFA Ct
KO + rescue MAFB	-DOX	28.38
KO + rescue MAFB	+DOX	27.36
		MAFB Ct
KO + rescue MAFB	-DOX	19.62
KO + rescue MAFB	+DOX	19.00
		GAPDH Ct
KO + rescue MAFB	-DOX	20.08
KO + rescue MAFB	+DOX	20.51

Response to reviewers - Figure 2. Representative qPCR results and Ct value for MAFB, MAFA and GAPDH in one independent experiment (above) and grouped results for MAFA and MAFB (below).

In the case of the rescue experiment, we see a trend of upregulation of MAFB in the DOX treated cells but a more pronounced increase in the protein levels as shown in **Figure 5A**.

- On Fig. 2, there is a 10-fold decrease in the number of C-Pept⁺ in MAFB^{-/-} cells. Based on Fig. 5, there is a qualitative trend to revert this phenotype upon MAFB reexpression, but it did not reach values obtained in control conditions. The authors may discuss this point.

This is likely due to the fact that MAFB is only expressed in one allele from AAVS1 locus in **Figure 5**. The relative amounts of MAFB are lower than in the MAFB^{+/+} and the partial rescue observed is more aligned with the MAFB^{+/-} phenotype. The amount of MAFB expressed in our +DOX cells is more aligned with the amounts of MAFB protein detected in the MAFB^{+/-} condition and thus we believe this indicates that dosage dependent effects of MAFB are relevant in this model.

We understand there are limitations to this approach that have now been discussed in the text.

- I am a bit puzzled by the biological relevance of the chapter “MAFB elimination compromises a-cell differentiation”. I read that in the in vitro model, “GCG mono-hormonal cells are achieved by transitioning from a previously noted C-PEP⁺ GCG⁺ co-expressing stage”. I thought that in vivo (at least

in the mouse), it is not the case. The difference between mouse and human or the limit of the in vitro system should be highlighted.

The reviewer is correct in pointing out that the formation of alpha cells progresses via different progenitors in rodent and human embryos. Notably, this polyhormonal cell state has previously been recognized in the human fetal pancreas (De Krijger et al., 1992). Through our beta-cell differentiation protocol, we are promoting the induction of NKX6.1 and Insulin and as a result, we also unintentionally appear to generate other hormone producing cell types such as Glucagon producing cells. This has previously been noted in the field and remains an active area of research as to how one can prevent or further differentiate cells past this poly-hormonal positive stage (Velazco-Cruz et al., 2019) (Rezania et al., 2014) (Rezania et al., 2011).

- Fig. 6G. CHGA levels are not changing, while INS and GCG mRNA are strongly decreased by more than 10 folds. I have the feeling that the increase in SST and PPY does not compensate the decrease in INS and GCG. Is there CHGA+ cells negative for the above endocrine markers?

We think this is an interesting point and agree that the number of SST+ and PPY+ cells likely does not fully compensate for the number of cells with a decrease in GCG and INS.

Our scRNA Seq data supports the hypothesis that there are CHGA+ hormone negative populations. Additionally, we also find increased expression of the hormones PYY and GAST in the MAFB-/- cells which might account for a number of the CHGA+ cells as they have been previously reported to be expressed in the developing mouse pancreas as well as in stem cell differentiation protocols (Suissa et al., 2013) (Upchurch et al., 1994). However, more work is required to assess the contribution of these hormones to endocrine cell specification in humans and in the context of MAFB loss.

- Fig. 6I: would be nice to have the data as absolute and not relative values.

The absolute values for GCG in MAFB +/+ and -/- cells is 0.26 ± 0.08 (pg)/ μ g DNA vs. 0.06 ± 0.01 (pg)/ μ g DNA and is now shown in **Figure 6I**.

There is a broad range of GCG basal secretion levels based on the individual differentiation experiment and thus we have now moved this to the supplemental data and show results per individual experiment (**Figure S15C**). The pattern is consistent that in transitioning from low glucose to high glucose we observe a reduction in the levels of GCG secreted in both MAFB +/+ and -/- cells. We plan to further investigate the response of MAFB-/- alpha cells to secretagogues in future studies.

Minor comments

- Introduction: is MAFB expressed in PP cells?

Unfortunately, we can't co-stain on human tissue as MAFB antibody and PPY are both antibodies are raised in rabbit. PPY expressing cells are relatively rare in the human pancreas but to date, no robust MAFB signal has been found in human PPY (γ) cells in published scRNA seq datasets relative to alpha, beta or delta cells.

Example data from Segerstolpe et al., is shown below for reference (Segerstolpe et al., 2016).

Visualizing MAFB expression across human pancreas:

Boxplots summarising MAFB expression levels across the 7 major cell types (shown in different colors) for every donor (shown in different shades of each color). The first 6 boxes correspond to healthy individuals (H1 to H6) and the last 4 to T2D individuals (T2D1 to T2D4). ϵ -cells were captured only in 5 donors (H2, H3, H6, T2D1, T2D4). **t-SNE representations** colored according to MAFB expression levels of: (Left) all sequenced cells (n=2,209) or (Right) endocrine cells only (n=1,554). Compare with the corresponding t-SNE graphs/maps shown below (reproduced from Figure 1 of the paper).

Reproducing Figs 1B and 1E of *Segerstolpe, Palasantza et al.* to show the placement of cell types onto the t-SNE maps.

Response to reviewers - Figure 3. Example of single cell RNA-seq from (Segerstolpe et al., 2016) showing very few γ -cells expressing MAFB.

- The right panel on Fig. 1C with MAFB is important but a bit difficult to read. It would be nice to deconstruct the panel with each individual marker.

We now show individual panels in Figure S1A.

- What about the long-term survival of EndoC-BH2 cells following MAFB knock-down? Do the cells continue growing?

Yes, the cells continue growing, as we expected due to the absence of MAFB expression in undifferentiated hESCs. There does not appear to be an impact on cell survival following MAFB knockdown. Cells were passaged for multiple weeks.

- Why did the authors measure pre-INS transcript in EndoC-BH2 and not INS transcripts?

We chose to measure pre-INS transcript because it is a more sensitive read-out of INS expression following relatively short-term/acute knockdown of MAFB. INS transcript is highly abundant and fairly stable. The primers and reasoning came out of this publication (Evans-Molina et al., 2007).

- Page 7 “or the b-like cell stage ($4.6 \pm 1.6\%$ vs. $5.7 \pm 2.4\%$) (Figure S4E,F).” It may be S5 instead of S4 We thank the reviewer for noticing this and the text has now been updated to indicate that this is now **Figure S9E,F**.

--

Reviewer #3 (Remarks to the Author):

In this manuscript, Russell et al. describe the role of transcription factor MAFB in differentiation of human pancreatic beta- and alpha-cells using hPSCs, iPSCs and EndoC-betaH2 cells, and demonstrate that MAFB regulates human endocrine cell lineage commitment. The loss of MAFB resulted in increased population of somatostatin (SST)- and pancreatic polypeptide (PPY)-expressing cells at the expense of glucagon (GCG)- and insulin (INS)-expressing cells. Induced expression of MAFB in MAFB^{-/-} cells partially rescued the phenotype above. Altered expression of transcription factors NKX6.1 and HHEX in addition to hormones was also demonstrated.

Results are clearly demonstrated with careful examination and generally convincing. However, many experiments are not described in sufficient detail and several important control data are missing. Additionally, results described in this manuscript are generally in line with published findings elucidated by analyses of genetically engineered mouse models (GEMMs) and are expected, although authors emphasize that their results are considerably different from those from GEMMs.

We thank the Reviewer for acknowledging the careful examination of the data and have updated the text to provide sufficient detail where it had been lacking. In line with comments from reviewer #1, we have re-worked the text to carefully highlight that much of the phenotype observed in MAFB^{-/-} cells is complementary to results described in GEMMs with MAFB loss. We do believe there are some discrepancies between the two models that we discuss in the text, however we also point out that additional work is required to expand on these observations. Overall, by incorporating the suggestions of the reviewer, we believe we have strengthened the manuscript considerably.

Major comments:

1) Fig. S1A-E shows only two results of MAFB^{+/+}, which is important as a control. Therefore, the experiments with MAFB^{+/+} hESC (and iPSCs) should be performed at least in triplicate as the minimum number.

This has now been updated. This experiment was performed using 3 independent differentiation experiments per clone and 2 independent WT clones used.

2) Please clarify how many mice and sections were examined for each immunofluorescence (IF) data, Figs. 1C, 1H, 2E, 3C, 4G, 4J, 5C, 5G and 6H, and show statistical analysis of these results. In addition, what kind of PPY antibody was used for IF? Is it specific to PPY?

FACS analysis is provided that provides quantitative data and representative IF images from at least 3 differentiation experiments are shown as reference.

Channels have been separated and are now shown throughout the supplemental data for space and brevity.

We received a recommendation of the PPY antibody from Al Powers group at Vanderbilt and performed validation on human pancreas sections. The sparse staining pattern and amount of these cells that are present indicate that this antibody specifically stains PPY producing cells in the pancreatic islet.

Response to reviewers - Figure 4. Representative images of PPY and GCG staining in human pancreas sections.

3) Results from GEMMs revealed that knockout of MAFB has not minimal but certain effect on murine beta- and alpha-cell development as described in references 7 and 8, which shows significantly reduced number of beta- and alpha-cells, while total number of endocrine cells was unchanged. It has also been described that number of somatostatin-expressing cells and ghrelin-expressing cells is unaffected in MAFB KO mice (Nishimura et al., Dev Biol 314, 443-456, 2008). Authors should discuss these previously reported results.

We have now updated the text to further highlight these previously published reports that MAFB KO in the mouse has effects on the number of alpha and beta cells in the developing pancreas. We find a similar pattern for these hormones in the context of human endocrine cell specification, but also highlight that we additionally observe significant increases in the number of SST and PPY hormone producing cell types. Overall, we acknowledge that the transcriptional network associated with pancreatic endocrine cell development is remarkably conserved and that subtle differences might be uncovered by utilizing human specific approaches in parallel to GEMMs.

4) Fig. 6I clearly shows that loss of MAFB resulted in reduced glucagon content and that glucose-regulated GCG secretion was similar between MAFB^{+/+} and MAFB^{-/-} cells. How are the INS content and secretion of MAFB^{+/+} and MAFB^{-/-} beta-like cells *in vitro*? Additionally, can SST induced in MAFB^{-/-} cells be secreted from cells at physiologically meaningful level?

As previously shown, our D20 cells are relatively immature and do not respond to glucose stimulation (Nair et al., 2019). While we performed these experiments, having no clear response in the wildtype setting prevented any interpretation of data from the MAFB^{-/-} cells. This is a fundamental reason for performing the *in vivo* experiment, which provide a microenvironment for maturation and functional assessment of cells at this stage. Notably, INS content *in vitro* for MAFB^{+/+}, ^{+/-} and ^{-/-} cells is outlined in **Figure S2I**.

We performed a glucose stimulation of D20 cells and identified that Somatostatin is secreted at very low levels from both the MAFB^{+/+} and ^{-/-} cells as shown below. Notably, there is a trend towards higher total SST levels in the MAFB^{-/-} cells as expected.

There is minimal data in the literature relating to SST release from human islets and their response to different stimuli such as fluctuating levels of glucose. Primarily, it has been shown that δ -like cells or even EndoC-BH2 cells respond to glucose by increasing SST secretion. However, we fail to find data showing dynamic regulation of this secretion by reducing SST in response to a shift back to low glucose as might be expected (Tsonkova et al., 2018) (Hauge-Evans et al., 2009).

We assessed SST and GCG levels in the same samples as those shown in **Figure 3** for C-PEP, however as these hormones are cross-reactive between mouse and human unlike C-PEP, robust conclusions were not possible due to background and thus, we do not include these data. We now have strategies in place for future studies to test their functionality *in vivo* that are beyond the scope of the current study.

[Redacted]

5) Regarding gene expression data of Fig. S2D and S2E (PP stage) and Fig. S4B (beta-like cell stage), please show expression of HHEX, MAFA, PAX4, ARX, PAX6, CHGA, and ALDH1a3, the dedifferentiation marker. To mention that “without affecting total numbers of endocrine cells as marked by CHGA expression” in page 7, line 8-9, please show evidence.

qPCR for these markers at the beta-like stage is now shown in **Figure S7B** and further discussed in the text.

The total numbers of CHGA as assessed by flow cytometry is outlined in **Figure 4D**.

Minor points:

6) In Fig. 3B, please show blood glucose data. How many times were islet transplantation experiments repeated?

We now include serum glucose levels demonstrating that the glucose IP bolus worked accordingly in all animals and that there is no significant difference in glucose levels detected between the MAFB +/+ and -/- transplanted animals (**Figure 3B**).

Transplantation experiments were performed one time with three independent NSG mice per group. There were no outliers and the data are consistent with the *in vitro* experiments. The figure legend has now been updated to reflect this.

7) Page 7, line6: “Figure 5C” should be “Figure 4C”. Page 7, line 8: “to compromise the composition of the endocrine cell composition” should be “to compromise the composition of the endocrine cells”.
The text has been updated.

8) Fig. 5A-D: It seems that number of INS-GFP cells in MAFB-/- cells with DOX-induced MAFB expression does not reach the level of those in MAFB+/+ cells. In Fig. 5A please show quantitative data of MAFB expression levels in MAFB+/+ cells together with MAFB-/-, iHYGRO-MAFB-DOX and iHYGRO-MAFB+DOX cells.

We now include a western blot showing expression of MAFB +/+, +/- and -/- cells on the same blot as iHYGRO-MAFB - or + DOX treated cells. These experiments were done in parallel with experiments shown in **Figure 2C** as reference.

References

Arda, H.E., Li, L., Tsai, J., Torre, E.A., Rosli, Y., Peiris, H., Spitale, R.C., Dai, C., Gu, X., Qu, K., *et al.* (2016). Age-Dependent Pancreatic Gene Regulation Reveals Mechanisms Governing Human beta Cell Function. *Cell Metab* 23, 909-920.

Bruin, J.E., Rezaia, A., Xu, J., Narayan, K., Fox, J.K., O'Neil, J.J., and Kieffer, T.J. (2013). Maturation and function of human embryonic stem cell-derived pancreatic progenitors in macroencapsulation devices following transplant into mice. *Diabetologia* 56, 1987-1998.

Cyphert, H.A., Walker, E.M., Hang, Y., Dhawan, S., Haliyur, R., Bonatakis, L., Avrahami, D., Brissova, M., Kaestner, K.H., Bhushan, A., *et al.* (2018). "Examining How the MAFB Transcription Factor Affects Islet β Cell Function Postnatally". *Diabetes*.

De Krijger, R.R., Aanstoot, H.J., Kranenburg, G., Reinhard, M., Visser, W.J., and Bruining, G.J. (1992). The midgestational human fetal pancreas contains cells coexpressing islet hormones. *Dev Biol* 153, 368-375.

Evans-Molina, C., Garmey, J.C., Ketchum, R., Brayman, K.L., Deng, S., and Mirmira, R.G. (2007). Glucose regulation of insulin gene transcription and pre-mRNA processing in human islets. *Diabetes* 56, 827-835.

Hang, Y., Yamamoto, T., Benninger, R.K., Brissova, M., Guo, M., Bush, W., Piston, D.W., Powers, A.C., Magnuson, M., Thurmond, D.C., *et al.* (2014). The MafA transcription factor becomes essential to islet beta-cells soon after birth. *Diabetes* 63, 1994-2005.

Hauge-Evans, A.C., King, A.J., Carmignac, D., Richardson, C.C., Robinson, I.C., Low, M.J., Christie, M.R., Persaud, S.J., and Jones, P.M. (2009). Somatostatin secreted by islet delta-cells fulfills multiple roles as a paracrine regulator of islet function. *Diabetes* 58, 403-411.

Nair, G.G., Liu, J.S., Russ, H.A., Tran, S., Saxton, M.S., Chen, R., Juang, C., Li, M.L., Nguyen, V.Q., Giacometti, S., *et al.* (2019). Recapitulating endocrine cell clustering in culture promotes maturation of human stem-cell-derived beta cells. *Nat Cell Biol* 21, 263-274.

Pasquali, L., Gaulton, K.J., Rodriguez-Segui, S.A., Mularoni, L., Miguel-Escalada, I., Akerman, I., Tena, J.J., Moran, I., Gomez-Marin, C., van de Bunt, M., *et al.* (2014). Pancreatic islet enhancer clusters enriched in type 2 diabetes risk-associated variants. *Nat Genet* 46, 136-143.

Rezaia, A., Bruin, J.E., Arora, P., Rubin, A., Batushansky, I., Asadi, A., O'Dwyer, S., Quiskamp, N., Mojibian, M., Albrecht, T., *et al.* (2014). Reversal of diabetes with insulin-producing cells derived in vitro from human pluripotent stem cells. *Nat Biotechnol* 32, 1121-1133.

Rezaia, A., Riedel, M.J., Wideman, R.D., Karanu, F., Ao, Z., Warnock, G.L., and Kieffer, T.J. (2011). Production of functional glucagon-secreting alpha-cells from human embryonic stem cells. *Diabetes* 60, 239-247.

Ricordi, C., Gray, D.W., Hering, B.J., Kaufman, D.B., Warnock, G.L., Kneteman, N.M., Lake, S.P., London, N.J., Socci, C., Alejandro, R., *et al.* (1990). Islet isolation assessment in man and large animals. *Acta Diabetol Lat* 27, 185-195.

Russ, H.A., Parent, A.V., Ringler, J.J., Hennings, T.G., Nair, G.G., Shveygert, M., Guo, T., Puri, S., Haataja, L., Cirulli, V., *et al.* (2015). Controlled induction of human pancreatic progenitors produces functional beta-like cells in vitro. *EMBO J* 34, 1759-1772.

Segerstolpe, A., Palasantza, A., Eliasson, P., Andersson, E.M., Andreasson, A.C., Sun, X., Picelli, S., Sabirsh, A., Clausen, M., Bjursell, M.K., *et al.* (2016). Single-Cell Transcriptome Profiling of Human Pancreatic Islets in Health and Type 2 Diabetes. *Cell Metab* 24, 593-607.

Suissa, Y., Magenheimer, J., Stolovich-Rain, M., Hija, A., Collombat, P., Mansouri, A., Sussel, L., Sosa-Pineda, B., McCracken, K., Wells, J.M., *et al.* (2013). Gastrin: a distinct fate of neurogenin3 positive progenitor cells in the embryonic pancreas. *PLoS One* 8, e70397.

Tsonkova, V.G., Sand, F.W., Wolf, X.A., Grunnet, L.G., Kirstine Ringgaard, A., Ingvorsen, C., Winkel, L., Kalisz, M., Dalgaard, K., Bruun, C., *et al.* (2018). The EndoC-betaH1 cell line is a valid

model of human beta cells and applicable for screenings to identify novel drug target candidates. *Mol Metab* **8**, 144-157.

Upchurch, B.H., Aponte, G.W., and Leiter, A.B. (1994). Expression of peptide YY in all four islet cell types in the developing mouse pancreas suggests a common peptide YY-producing progenitor. *Development* **120**, 245-252.

Velazco-Cruz, L., Song, J., Maxwell, K.G., Goedegebuure, M.M., Augsornworawat, P., Hoglebe, N.J., and Millman, J.R. (2019). Acquisition of Dynamic Function in Human Stem Cell-Derived beta Cells. *Stem Cell Reports* **12**, 351-365.

Wesolowska-Andersen, A., Jensen, R.R., Alcantara, M.P., Beer, N.L., Duff, C., Nylander, V., Gosden, M., Witty, L., Bowden, R., McCarthy, M.I., *et al.* (2020). Analysis of Differentiation Protocols Defines a Common Pancreatic Progenitor Molecular Signature and Guides Refinement of Endocrine Differentiation. *Stem Cell Reports* **14**, 138-153.

Reviewers' Comments:

Reviewer #1:

Remarks to the Author:

Authors have addressed comments satisfactorily

Reviewer #2:

Remarks to the Author:

The authors properly took into account my comments that aimed at further enhancing the quality of the manuscript

Raphael Scharfmann

Reviewer #3:

Remarks to the Author:

Overall, authors addressed questions raised by reviewers and revised the manuscript as suggested. Although authors mentioned that immunofluorescence images result from at least three biological replicates and are quantitative with FACS data, this reviewer still has concern that each experiment is not described in sufficient detail in current manuscript. Authors should clarify how many times each experiment was performed by showing the number of replicates in the figure legends. Also as author replied, difference in transcription network and mechanism of differentiation between mice and human is subtle, and insulin-, glucagon and somatostatin-producing cells in the embryonic pancreas of MafB knockout mice were already analyzed and discussed as this reviewer indicated in previous comments #3. Therefore, authors should revise the statements such as "knockout of MAFB has minimal effect on murine beta-cell development and function" or "uncovers previously unappreciated roles in other pancreatic endocrine cell types, in particular delta-cells" in the summary.

REVIEWERS' COMMENTS:

Reviewer #1 (Remarks to the Author):

Authors have addressed comments satisfactorily

We thank the reviewer for their appreciation of our work.

Reviewer #2 (Remarks to the Author):

The authors properly took into accounts my comments that aimed at further enhancing the quality of the manuscript

Raphael Scharfmann

We thank the reviewer for their constructive comments and approval of our work.

Reviewer #3 (Remarks to the Author):

Overall, authors addressed questions raised by reviewers and revised the manuscript as suggested. Although authors mentioned that immunofluorescence images result from at least three biological replicates and are quantitative with FACS data, this reviewer still has concern that each experiment is not described in sufficient detail in current manuscript. Authors should clarify how many times each experiment was performed by showing the number of replicates in the figure legends. Also as author replied, difference in transcription network and mechanism of differentiation between mice and human is subtle, and insulin-, glucagon and somatostatin-producing cells in the embryonic pancreas of MafB knockout mice were already analyzed and discussed as this reviewer indicated in previous comments #3. Therefore, authors should revise the statements such as “knockout of MAFB has minimal effect on murine beta-cell development and function” or “uncovered previously unappreciated roles in other pancreatic endocrine cell types, in particular delta-cells” in the summary.

We thank the reviewer for their appreciation of our work. We have revised the figure legends as suggested to clarify how many times each experiment was performed by including the number of replicates in the figure legends throughout the manuscript. In addition, we have updated the summary to comply with the 150 word limit and removed the statements causing concern to this reviewer. Notably, an increase in SST positive delta-cells has never been reported in MAFB KO mouse models in contrast to what we observe in this human specific context.